# SPATIALLY-AWARE TRANSFORMER FOR EMBODIED AGENTS

**Junmo Cho**[1*], **Jaesik Yoon**[1,2*], **Sungjin Ahn**[1]
[1]KAIST & [2]SAP

## ABSTRACT

Episodic memory plays a crucial role in various cognitive processes, such as the ability to mentally recall past events. While cognitive science emphasizes the significance of spatial context in the formation and retrieval of episodic memory, the current primary approach to implementing episodic memory in AI systems is through transformers that store temporally ordered experiences, which overlooks the spatial dimension. As a result, it is unclear how the underlying structure could be extended to incorporate the spatial axis beyond temporal order alone and thereby what benefits can be obtained. To address this, this paper explores the use of Spatially-Aware Transformer models that incorporate spatial information. These models enable the creation of place-centric episodic memory that considers both temporal and spatial dimensions. Adopting this approach, we demonstrate that memory utilization efficiency can be improved, leading to enhanced accuracy in various place-centric downstream tasks. Additionally, we propose the Adaptive Memory Allocator, a memory management method based on reinforcement learning that aims to optimize efficiency of memory utilization. Our experiments demonstrate the advantages of our proposed model in various environments and across multiple downstream tasks, including prediction, generation, reasoning, and reinforcement learning. The source code for our models and experiments will be available at https://github.com/junmokane/spatially-aware-transformer.

## 1 INTRODUCTION

Episodic memory is the type of memory that involves storing and retrieving events or episodes from an individual's life. This memory plays a crucial role in various human cognitive processes, such as enabling people to mentally travel back in time and remember past experiences like "what you did yesterday" (Perrin & Michaelian, 2017; Suddendorf et al., 2009). It is also foundational in imagining the future and making effective decisions (Schacter et al., 2017). Therefore, AI systems aiming to mimic human-like cognitive abilities would benefit from incorporating a form of episodic memory.

Currently, the primary approach for modeling episodic memory in AI is through transformers (Parisotto et al., 2020; Lampinen et al., 2021). This approach represents an episodic memory as a sequence of experience frames. During the storing process, frames are stored in First-In-First-Out (FIFO) order with temporal-order embedding, called positional embsedding. While this FIFO inductive bias seems reasonable in modeling episodic memory due to the uni-directional flow of time in the physical world, various studies in cognitive science suggest that the formation and retrieval of episodic memories rely not only on the temporal aspects but also on another foundational axis of the physical world, the spatial axis, such as where we parked our car this morning (Buzsáki & Tingley, 2018; Ekstrom & Ranganath, 2018; Pathman et al., 2018; van Asselen et al., 2006).

Despite this significance, the spatial dimension has largely been overlooked in the modeling of transformer-based episodic memory. This can be attributed to two main reasons. First, since their invention, transformer models have mainly advanced in the language domain where spatial dimensions are not as prominent or meaningful. Second, in some domains, spatial annotations, such as the position of a robot, may not be as easily accessible as the temporal order indices. As a result, previous research have concentrated mainly on improving the accuracy of inferring such spatial

---

*Equal contribution. Correspondence to Sungjin Ahn <sjn.ahn@gmail.com>.

information from experience, e.g., sequences of observations and actions (Fraccaro et al., 2018; Uria et al., 2020; Gregor et al., 2019; Whittington et al., 2022).

However, beyond language, there are a broad range of domains, such as robotics and virtual agents in games, where the spatial dimension is just as natural and crucial as the temporal dimension. These agents need to navigate around rooms and various areas, and having spatially structured knowledge such as "*what happened where*" is essential for achieving the goal. Moreover, in many cases, the spatial information is as easily accessible as the time indices. In virtual worlds such as games, for instance, the ground-truth position information is easily provided from the game engine (Guss et al., 2019) and for robots, it is already popular to provide such spatial information by a separate hardware system such as BLE beacons (if it is in-door) (Faragher & Harle, 2015), GPSs (El-Rabbany, 2002), or algorithmically, e.g., SLAM (Taketomi et al., 2017; Zhang et al., 2017). Despite these circumstances, transformers currently only consider time axis.

In this work, we pose the question: *what if spatial information is available as time indices to transformer-based models?* Our goal is to investigate this overlooked aspect of transformers and to examine whether and how transformers can utilize the spatial axis. We approach this by deviating from the traditional focus on acquiring spatial information. Considering the widespread use of transformer architectures, as well as the growing accessibility of spatial information for embodied agents, we believe it is appropriate and timely to address this question.

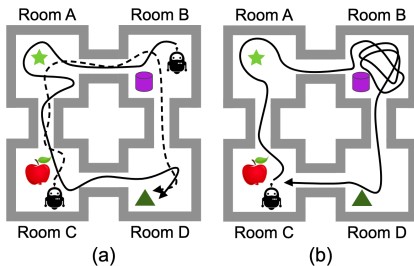

To achieve this goal, we propose a series of transformer architectures called Spatially-Aware Transformers (SAT or SA-Transformer), which incorporate spatial information into transformers. This allows us to explore various feasible design choices and assess the pros and cons of these architectures. Specifically, we demonstrate that Spatially-Aware Transformers offer three main benefits. Firstly, they **enhance spatial reasoning** tasks. We show that without explicit spatial information (only with temporal information), it is challenging to solve various general spatial reasoning tasks, such as "What has happened in the next room of the agent?" (as shown in Figure 1 (a)). However, providing spatial information makes these tasks much easier. Secondly, they enable **efficient memory management**. By incorporating spatial information, we can overcome the limitations of the FIFO memory, such as deleting important memory solely because it is old (as shown in Figure 1 (b)). We can also implement structured approaches like place-centric hierarchical memory. We demonstrate that this structured memory improves effectiveness and efficiency in various place-centric tasks. To enhance memory utility further, we introduce the Adaptive Memory Allocator (AMA). Unlike traditional transformers that follows the FIFO policy for memory writing, AMA allows for adaptive selection of various writing strategies based on the goal. Finally, we demonstrate that these **benefits extend** across a wide range of machine learning problems, including supervised prediction, image generation, and reinforcement learning.

**Figure 1:** Home robot thought experiments. In scenario (a), a robot visits the environment in a different temporal order. When it arrives at Room D, answering what happened to the room on its left would not be easy using only the temporal axis. In scenario (b), the robot stays in Room B for a long time. With a FIFO memory, the agent would delete the memory of Room C. When it returns, it will perceive Room C as new.

The main contributions of the paper are as follows: First, to our knowledge, we are the first to motivate, conceptualize, and introduce the notion of transformers capable of utilizing explicit spatial information. Second, we propose a series of SAT models designed to leverage spatial annotation. Third, we introduce the AMA method, which enables adaptable selection of various memory management strategies beyond the traditional FIFO-only strategy. Finally, we demonstrate that the advantages of SAT are not limited to a specific problem domain but extend across an array of machine learning applications, including supervised prediction, image generation, and reinforcement learning. We also release the source code for reproducibility.

## 2 PROPOSED MODELS

In this section, we describe the proposed Spatially-Aware Transformer (SAT) models. We begin with the simplest model, SAT-FIFO, and then discuss incorporating Place-Memory into SAT. Lastly, we

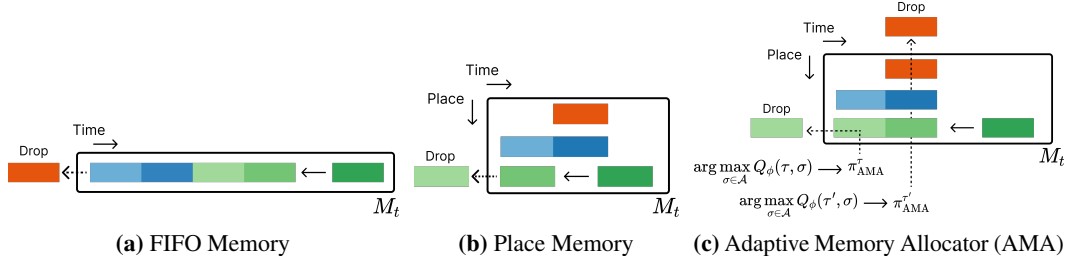

**Figure 2:** Illustrations of FIFO, Place Memory, and the Adaptive Memory Allocator (AMA). The orange memory is the oldest and originates from a different place compared to the green and blue ones. In FIFO, the orange memory is dropped due to its age. However, the Place Memory system retains it through place-wise memory allocation. AMA chooses the appropriate allocation strategy, $\pi_{\text{AMA}}^\tau$, based on the given downstream task knowledge, denoted as $\tau$.

introduce Adaptive Memory Allocation as a method for selecting a strategy other than FIFO based on the purpose of the downstream tasks.

## 2.1 DESIGN 1: SAT WITH FIFO MEMORY

When location annotation $\ell_t$ is available at each time step $t$, the most direct approach to make a spatially-aware episodic memory is simply to introduce the spatial embedding $e_t^{loc} = \texttt{embed}(\ell_t)$ and combine it with the time embedding[1] $e_t^{time}$ and the observation embedding $e_t^{obs}$. For the time and spatial embedding, the learnable embedding (Devlin et al., 2018) or sinusoidal positional embedding (Vaswani et al., 2017) could be applicable, but in this literature, for simplicity, the sinusoidal positional embedding is used. This results in the experience frame $x_t = \texttt{sum\_embed}_x(e_t^{loc}, e_t^{time}, e_t^{obs})$. Then, we add the experience frame to the episodic memory $M_t$ in the conventional FIFO order. We call this simplest model SAT-FIFO and its memory structure is illustrated in Figure 2a.

The main advantage of this approach is that it requires minimal alteration from the standard FIFO transformer in implementing the three basic operations of an episodic memory: WRITE ($M_{t+1} = \texttt{Write}(M_t, x_t)$), READ ($y_t = \texttt{Read}(M_t, q_t)$), and REMOVE ($M_{t+1} = \texttt{Remove}(M_t, \sigma)$). Here, $q_t$ represents the query used to retrieve information from the episodic memory, and $y_t$ is the output of the retrieval operation that will be used to solve downstream tasks. When it comes to removal, we generalize the operation by introducing a strategy $\sigma$ that determines how to remove experience frames. In the case of SAT-FIFO, the strategy $\sigma$ is fixed to FIFO. Although this approach endows the standard transformer with spatial awareness with minimal effort, it has limitations. For example, when the memory is full, the SAT-FIFO model has to remove the oldest experience first even if it is important to keep it for achieving the goal of the downstream task, e.g., as illustrated in Figure 1 (b).

## 2.2 DESIGN 2: SAT WITH PLACE MEMORY

To resolve the limitations of SAT-FIFO, we introduce the Place Memory (PM) into the Spatially-Aware Transformers. In this model, called SAT-PM, the whole space is represented by a number of places and we maintain a place-wise episodic memory $M_t^k$ one per place $k = 1, ..., K$, resulting in a place-centric hierarchical episodic memory $M_t = \{M_t^k\}_{k=1}^K$, as shown in Figure 2b.

A place can be viewed as a cluster of locations and we observe that this place structure is naturally provided in many applications, e.g., via BLE beacons for indoor robots, via GPS and maps for autonomous driving cars, and via clustering algorithms in general. Even if the place structure must be learned through a clustering algorithm, we note that it is a relatively simple learning task due to the extremely low dimensionality (only 2 or 3) of the spatial coordinates. Moreover, as we shall show in Exp-5 in Sec. 3.2, it is not always necessary to have the ground truth place structure but an approximate one can be adequate in practice. Thus, in our experiments, we assume that the location is already clustered and thus replace the location embedding with place embedding $e_t^{\text{place}}$.

**Place-Centric Store:** To store the experience frame $x_t^k$ obtained at place $k$, it is added to $M_t^k$ in a FIFO order. The length of each place's episodic memory, $L_p$, can be set to $L/K$ by default or

---

[1]Technically, it is not time embedding, but *position* (or temporal order) embedding. However, for simplicity, we refer to it as time embedding throughout the paper because "position" can also refer to spatial position.

manually adjusted with the prior's guidance. This allows the Place Memory to retain the memory of a room, regardless of how long ago it was visited, as long as the place's memory is not full. Unlike the flat FIFO transformer, this approach provides greater flexibility, balancing total memory capacity between time and space and help mitigate the issue shown in scenario (b) in Figure 1. However, if a place's memory is full, experience frames will still be removed in a FIFO order. Although such deletions are inevitable in any episodic memory with finite capacity, in the next subsection, we will discuss different memory management strategies that can handle this situation better than FIFO.

**Place-Centric Hierarchical Read:** A critical issue with transformers is their computation cost increasing with the number of memory items (Vaswani et al., 2017). One approach to mitigate this problem in an episodic transformer is to group a set of experience frames (Lampinen et al., 2021) into *chunks* and perform hierarchical read operations. Specifically, we first select the top-k chunks by applying attention only across the chunk representations instead of individual experience frames. Then, we search for experience frames within those chosen chunks, as shown in Figure 9.

Previously, only time-based chunking was available. However, we observe that this approach can lead to problems in spatial environments. This is because experience frames of a place can be scattered across chunks located at various positions in the flat memory (depending on the visit order of a trajectory), and similarly, a chunk can contain mixed experiences from multiple places. This makes it more difficult to learn to search for these frames and develop a representation of what happened in a place. Utilizing spatial awareness, our Spatially-Aware Transformer provides a simple solution to address this issue. We first ensure that a chunk is filled only with experiences of the same place, and these chunks are managed within a place memory. In experiments, we show that this simple place-centric hierarchical reading is both fast and effective for spatial reasoning. For more details about the hierarchical operation, please refer to the Appendix B.2.

## 2.3    DESIGN 3: ADAPTIVE MEMORY ALLOCATOR

So far, we have assumed that experience frames are added to each place memory in the FIFO order. While using the inductive bias, "*recent knowledge is more important*", is reasonable in modeling languages, it may not hold for various spatial problems. For example, a task may simply require retaining memory of a place that occurred at the beginning of an episode. Hence, to have a generic episodic memory, a more desired approach is to decide what to remember or forget *adaptively* according to the goal of the task.

However, a dilemma exists in practice. If we prioritize flexibility, the best approach would likely be to learn the optimal decision of which knowledge to be updated *at each time step* (Graves et al., 2014; 2016; Hung et al., 2019). However, it is known that this approach is notoriously difficult to train in practice, and we show it empirically in Appendix D.6. In fact, transformers completely remove the problem of learning to write into the memory by adopting the FIFO policy. This is one of the most crucial advantages of the transformer model, which allows it to be scalable in the language domain but at the cost of losing adaptability to find a better writing strategy.

To address this issue, we propose a simple and practical method called Adaptive Memory Allocator (AMA) that balances flexibility of the writing policy and ease of training. AMA extends the space of memory management strategies to multiple strategies by allowing the developer to design the space of various memory management strategies, denoted as $\mathcal{A} = \{\sigma_1, \ldots, \sigma_M\}$, based on their prior knowledge. For instance, $\sigma_1$ can be FIFO, $\sigma_2$ can be Last-In-First-Out (LIFO), $\sigma_3$ can be Least-Visited-First-Out (LVFO), and so on. This way, the traditional transformer can be viewed as a special case where there is only one strategy, i.e., FIFO. Also, these strategies are similar to CPU-cache management algorithms (Denning, 2005), but our goal is to learn how to choose between them. It is important to note that the primary value of this method lies not in what the strategies themselves are, as they can be created in various ways based on prior knowledge. Rather, the value lies in the fact that Spatially-Aware transformers enable the selection of these strategies (other than FIFO) in a way that optimizes downstream tasks.

The problem is then formulated as choosing the best strategy based on the agent's current intention or task description $\tau$. For example, the agent can be given $\tau =$ "*What object did you see in your left room?*" This problem can be addressed by learning a policy $\pi(\sigma|\tau)$. We implement this as a one-step Q-learning policy $\pi_{\text{AMA}} = \arg\max_{\sigma \in \mathcal{A}} Q_\phi(\tau, \sigma)$, where $\phi$ is the parameter of the value function. One might think that, if the number of strategies is small enough, we can simply try all strategies and

pick the best one. Then, we do not need to learn how to choose one. However, as there are almost infinitely many ways of describing a task, e.g., via language, it should be able to generalize to the variations of task descriptions. By learning the context-aware policy in AMA, we can learn to choose a suitable strategy even if an unseen (as a variation of) task description is given at test time. AMA is illustrated in Figure 2c, and the pseudo-code for learning algorithm with SAT and AMA is presented in the Appendix B.3.

# 3 EXPERIMENTS

This section presents an empirical evaluation of the Spatially-Aware Transformer (SAT) and Adaptive Memory Allocator (AMA) across various environments and downstream tasks. We begin by evaluating the models on prediction tasks in the Room Ballet environment. Then, we demonstrate its capability to build an action-conditioned world model and spatially-aware image generation model. Lastly, we showcase the use of SAT-AMA for training a downstream reinforcement learning policy. Each baselines and tasks are explained in each of the experiment section. In the experiments, observation, the time-step, and space are sequentially presented to the agent. Afterward, a query is provided. For some experiments, the task type is introduced at the episode's commencement to select the proper memory allocation strategy. The agent is expected to integrate these inputs to solve the tasks. The more details are in Appendix C.1.

## 3.1 SUPERVISED PREDICTION WITH ROOM BALLET

In this section, we use the Room Ballet environment, inspired by the Ballet task described in (Lampinen et al., 2021). Figure 3 illustrates the environment, which consists of multiple rooms. By default, there are 9 rooms, but this can be expanded to 16 and 25 in Exp-2. Each room represents a "place" and contains a dancer. In each episode, the agent is placed randomly in one of the rooms and moves through the rooms via random walk. Upon entering a room, the agent observes a complete dance performance consisting of 32 frames, performed by the dancer in that room. In each episode, a *dancer type* (i.e., the appearance of a dancer) and a *dance type* (i.e., the performance) are randomly and independently assigned to each room.

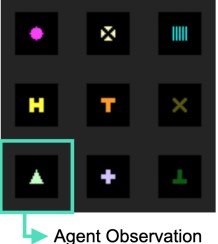

**Figure 3:** The Room Ballet environment.

**Exp-1. Implicit derivation of spatial information in transformers**

To verify that the transformers can derive the spatial information, we designed a spatial reasoning task on the Room Ballet environment, which we call the Next Ballet task. In this task, the agent visits 40 rooms (1280 steps) through a random walk with the four actions. Afterward, when provided with a query image showing a dancer's appearance, it is tasked to predict the type of dance that was performed in *the next room of the queried dancer* in clockwise order.

Although transformers have demonstrated impressive performance across various applications (Brown et al., 2020; Dosovitskiy et al., 2020; He et al., 2022; Parisotto et al., 2020; Micheli et al., 2022), it is unclear whether they can infer spatial information from navigation trajectories consisting of observations, actions, and time. To investigate this, we evaluate the following models: a standard transformer only with temporal order information T-FIFO, a transformer with temporal order and action T-FIFO+A, and a transformer with spatial information SAT-FIFO. Also, to ensure that memory capacity does not affect the results, we set the memory size to be large enough to store all observations.

As shown in the Figure 4 (a), we can see that the transformer using only temporal order without actions (T-FIFO) completely fails on this spatial reasoning task, suggesting its inability to develop an internal representation of the space from the ordered sequence of observations, as expected in the Thought Experiment. When we additionally provide the action information (T-FIFO+A), it began to show a better performance but still significantly inadequate to accurately solve the spatial reasoning problem by inferring the dancer's room location. These results suggest that inferring spatial information from observations and actions is not trivial in practice, though it may not be totally impossible. On the contrary, explicitly incorporating spatial information, SAT successfully solves the task, demonstrating the importance of having explicit spatial information in spatial reasoning tasks.

**Exp-2. Place-centric hierarchical reading vs Time-centric hierarchical reading**

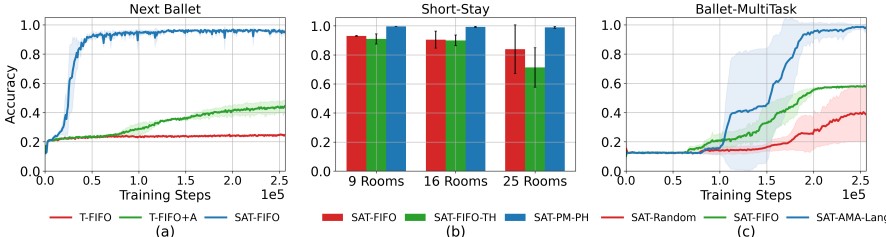

**Figure 4:** (a) The performance comparison of transformer memories equipped with different embeddings on the spatial reasoning task. (b) The performance comparison of unstructured memory and temporally/spatially structured memory on a spatial reasoning task in a large environment. (c) The performance comparison of the AMA model and predefined memory allocation strategies on a spatial reasoning task.

To demonstrate that place-centric chunking proposed in Section 2.2, is more beneficial for spatial task compared to traditional time-based chunking, we introduce a new and more realistic task called "Short Stay" in the Ballet environment. In this task, unlike the previous one, the agent is not required to stay in a room until it observes the entire dance. Instead, the agent exhibits more natural behavior by being able to move to a neighboring room at any time step. After making 1280 such moves, the agent is presented with a query dancer image and tasked with predicting its dance type. To verify the efficacy of the place-centric hierarchical SAT, we compared three models: (i) SAT-FIFO, (ii) SAT-FIFO with time-centric hierarchical read (SAT-FIFO-TH), and (iii) SAT-PM with place-centric hierarchical read (SAT-PM-PH). The accuracies are measured after 200K training steps, and the chunk size is set to 32. More details about this task are provided in Appendix C.3.

As shown in Figure 4 (b), it is evident that SAT-PM-PH consistently outperforms SAT-FIFO-TH and SAT-FIFO in all scenarios. Furthermore, as the number of rooms increases, the performance gap between SAT-FIFO-TH and SAT-PM-PH also widens. We observe that this is because SAT-FIFO-TH allows for temporally consecutive experiences from different places to be grouped together within a chunk. Similarly, experiences from the same place can be spread across multiple chunks. Consequently, gathering information about an event that occurred in a specific location by collecting scattered experience frames across multiple chunks remains a more challenging task for SAT-FIFO-TH, despite its use of place encoding and computational speed enhancement due to the hierarchy. On the other hand, SAT-PM-PH easily overcomes this challenge since a chunk contains only experiences from the same place. When compared to the non-hierarchical SAT-FIFO, SAT-PM-PH not only achieves better accuracy but also demonstrates faster inference speed, as in Table 1 in Appendix. This is because SAT-PM-PH only attends to the memory in the top-$k$ relevant chunks, while SAT-FIFO needs to attend to the entire memory.

**Exp-3. Learning to select memory allocation strategy with AMA**

In this section, we assume that the memory capacity is limited and demonstrate that with SAT-AMA, it is possible to learn to find a proper memory management strategy that best fits the downstream task and that it is better than always using FIFO. We first introduce four strategies: First-In-First-Out (FIFO), Last-In-First-Out (LIFO), Most-Visited-First-Out (MVFO), and Least-Visited-First-Out (LVFO). FIFO/LIFO replace the oldest/newest observation in the memory. MVFO/LVFO replace the oldest memory from the most/least frequently visited room. Note that whether these strategies are good or not is not the primary value of this work. Rather, they are examples of prior strategies that can be designed.

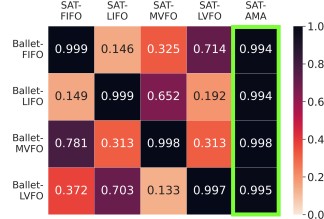

**Figure 5:** The accuracy of different task-strategy pairs.

Then, we introduce four different tasks, one for each strategy: Ballet-FIFO, Ballet-LIFO, Ballet-MVFO, and Ballet-LVFO. In the tasks, which consist of a total of 9 rooms as shown in Figure 3, the agent changes rooms 17 times (for a total of 576 steps) using random walk. This allows the agent to visit a room multiple times, potentially encountering multiple distinct dancers in a room. It is important to note that we limit the memory capacity to 288 and thus a half of the total episode steps must be chosen to be removed according to the management strategy. We note that the agent changes rooms after observing a complete dance performance in those tasks.

In each task, the agent is asked to predict the dance type of a queried dancer image. The selection of the query dancer follows a specific strategy, ensuring that it can only be answered correctly if the

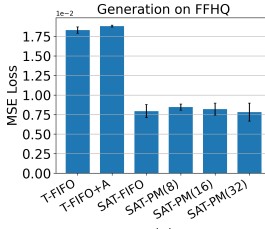 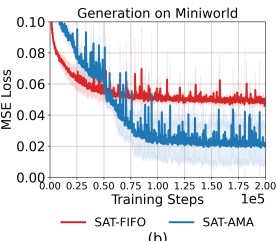 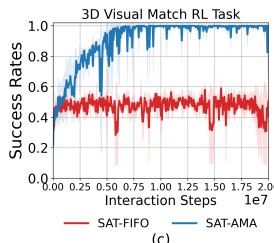

**Figure 6:** (a) The generation performance comparison for the navigation task on facial images from the FFHQ dataset. For SAT-PM, the numbers in parentheses are the number of the approximated places. (b) The generation performance comparison on the MiniWorld environment, which is a 3D non-grid environment. (c) The performance on the Visual Match task on the MiniWorld environment.

correct storage strategy was used. For instance, in the Ballet-LIFO task, the query dancer is always selected from the oldest half of the episode. Therefore, if the FIFO strategy was used, the memory does not contain the oldest half, making it impossible to correctly answer the query. However, if LIFO is chosen, the memory always retains the necessary experience to answer the query. Similarly, in the Ballet-MVFO/LVFO tasks, the agent is asked to predict the most recently observed dances within each room and the dances in the most frequently visited room, respectively. Importantly, given a task, the agent does not know what the best strategy is but needs to learn to choose it via AMA. Note that we use place-centric hierarchical read for selecting MVFO/LVFO, and time-centric hierarchical read for selecting in FIFO/LIFO in AMA. More details about the tasks are provided in Appendix C.4.

Figure 5 depicts the performance of different task-strategy pairs. As anticipated, we observe strong performance in the diagonal terms of the matrix when the strategy aligns with the task. Conversely, when the strategy does not match (i.e., the off-diagonal terms), the performance deteriorates. Notably, in the 5th column, we present the performance of SAT-AMA. SAT-AMA achieves comparable performance to strategies specifically designed for each task (the diagonal terms) as it learns to discover suitable strategies for each task.

**Exp-4. Generalization of SAT-AMA to multi-task descriptions**

In Exp-3, we demonstrated that SAT-AMA can learn to select an appropriate strategy for a single task. However, in this specific scenario, it is possible to find the correct strategy by trying out all configurations, as there are only four tasks and four strategies. In a more realistic scenario, it is common to ask the agent to solve multiple tasks with diverse formats, such as language descriptions (as described in Sec. 2.3). In such cases, the model should be able to generalize to the diverse variations of task descriptions. In this setting, it is impossible to try all possible configurations.

To test this, we mapped each task in Exp-3 to a set of natural language descriptions. We created 20 different descriptions for each task. For example, for the Ballet-LVFO task, we provided descriptions such as "*Recall the dancers in the room the agent frequented the most*" or "*Reflect on the dancers in the room that seemed to be the agent's favorite*". During training, we randomly selected 64 mixed descriptions (16 per task) and used the remaining 16 descriptions (4 per task) to evaluate SAT with AMA. We name this version of SAT as SAT-AMA-Lang. All descriptions used in the experiment can be found in Appendix C.5. For AMA, the description is given as an embedding from the OpenAI embeddings API [2]. The results are shown in Figure 4 (c). It is important to note that the accuracy in the evaluation is an average of three different training/evaluation description sets. In this case, SAT-AMA achieved an accuracy of over 95%. This indicates that AMA can be extended to generalize to the diverse variations of task descriptions and has potential in broad applications such as language-instructed robots.

### 3.2 EPISODIC IMAGINATION VIA IMAGE GENERATION

In this section, we demonstrate the potential of applying SAT to more complex tasks and environments: action-conditioned episodic image generation requiring multi-step spatial reasoning, and episodic image generation with AMA.

**Exp-5. Action-conditioned generation in the FFHQ world**

---

[2]We took the embedding from the model text-embedding-ada-002. https://platform.openai.com/docs/guides /embeddings/what-are-embeddings

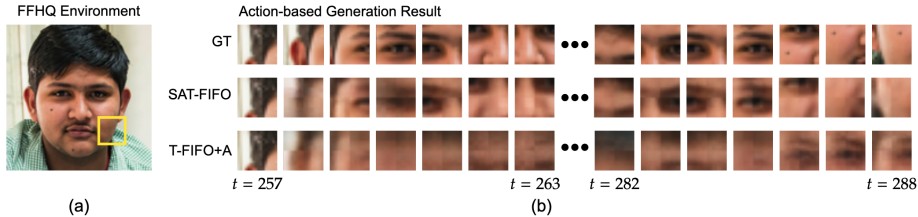

**Figure 7:** (a) FFHQ environment. The agent observes the partial view (yellow box) in the face image from FFHQ dataset. (b) Action-based generation result of SAT-FIFO and T-FIFO+A after Observation phase.

We demonstrate the capability of the SAT to generate conditioning on actions while navigating an environment to see its potential to implement an episodic world model (Gregor et al., 2019; Hafner et al., 2020; Micheli et al., 2022; Chen et al., 2022). To test this, we designed a navigation task on the facial images from FFHQ dataset (Karras et al., 2019), where 62,000 face images are used for training and 7,000 images among the remaining images are used for evaluation. The environment is a face image which is considered as a $10 \times 10$ grid map. The agent observes a partial image of the face, as shown in Figure 7 (a), while navigating the image provided with an action (up, down, left, right) at each step. During the observation phase, the agent spawns in a random location and performs a random walk for 256 steps. During the generation phase, at a random position, it is given a sequence of random actions of length 32. Then, the task is to generate images corresponding to the queried action sequence by utilizing the episodic memory constructed during the observation phase. Therefore, this task requires multi-step spatial reasoning. See Figure 17 in Appendix for more visualization. For this task, 2-D sinusoidal positional embedding that encodes x and y axes separately, is utilized for the space embedding. The comparative analysis of the embedding types for the space embedding is discussed in Appendix D.9.

To verify the efficacy of the SAT-FIFO, we compared it with T-FIFO and T-FIFO+A. The results are shown in Figure 6 (a) and 7 (b). Since this task is a harder version of the Next Ballet task in Exp-1 due to a larger map, T-FIFO and T-FIFO+A failed to solve it. SAT-FIFO, on the other hand, outperformed them and generated high-quality images. We note that this result is from the evaluation set which contains facial images that are unseen during training. This result suggests that SAT can handle complex spatial reasoning on unseen visual environments. Additionally, we investigated the effect of using approximate place structures by changing the number of clusters ($N = 8, 16, 32$). As shown in 6 (a), SAT-PM($N$) showed robust performance across different place clustering. See Appendix D.7 for more results and discussion.

**Exp-6. Image generation in 3D first-person view environment**

Next, we evaluate SAT with AMA for the image generation in the MiniWorld environment (Chevalier-Boisvert, 2018), a 3D environment where an agent navigates in the first-person view setting. As shown in Figure 11 in the Appendix, the agent is placed at the center of a room with four dancers positioned at each side of the room (North, South, East, and West), each performing a distinct dance. The agent observes the room by rotating at a random degree between 3 and 7 in a clockwise order. Afterwards, the agent is tasked with generating a dance video featuring the queried dancer. Due to the rotation, during the observation phase, the agent observes the dance from various viewpoints, as shown in Figure 11. However, during the generation phase, the agent is asked to generate the dance from the front view. Therefore, the model cannot simply copy the dance from its episodic memory, but instead needs to develop a proper representation of the dance from the episodic memory and decode it accordingly. Similar to the Ballet tasks, we tested four different types of queries: FIFO, LIFO, MVFO, and LVFO. As depicted in Figures 6 (b) and 11, we can see that SAT-AMA successfully learns to choose the appropriate strategy for the given task. In contrast, SAT-FIFO performs worse because it loses necessary memory if it is older than the memory capacity. More details about the experiment are provided in Appendix C.6.

## 3.3 REINFORCEMENT LEARNING AGENT

Lastly, we test an RL agent equipped with SAT-AMA memory. The task involves two rooms. The agent is initially placed in a room with a single box of a random color (either blue or green). After staying four steps in the room, the agent is transported to the second room which is larger and contains multiple yellow boxes, and navigates for 80 steps.

After this observation phase, the agent is warped back to the first room, where it is tasked to collect the box that matches the color of the box observed initially. The memory capacity is limited to 40, a half of the number of total steps in the second room. We use Proximal Policy Optimization (PPO) (Schulman et al., 2017) for agent learning. AMA is designed to learn through the reward from the environment and to choose a strategy between FIFO and MVFO. As shown in Figure 6 (c), SAT-AMA successfully learned to select the appropriate strategy (MVFO) and solve the task. While it only needs to choose one out of two strategies, it is important to note that this still a challenging task where AMA needs to learn through sparse rewards and many interaction steps, and in the first-person view 3D environment.

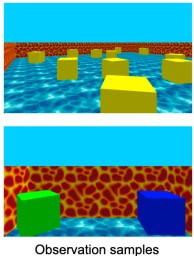

Observation samples

**Figure 8:** RL environment visualization.

## 4 Related Works

Transformers have been widely used in various applications (Brown et al., 2020; He et al., 2022; Parisotto et al., 2020; Chen et al., 2021), however, their high computational cost has been a major drawback, especially for reinforcement learning (Parisotto & Salakhutdinov, 2021). To address this issue, several methods have been proposed to reduce the computational cost of transformers without sacrificing their long-term memorization ability (Parisotto & Salakhutdinov, 2021; Lampinen et al., 2021). In this paper, we propose a new type of hierarchical read operation that is more efficient for place-centric tasks. We also propose a memory management method based on reinforcement learning that aims to optimize memory usage.

In parallel, there has been growing interest in using spatial information for embodied agents (Zhang et al., 2017; Rosenbaum et al., 2018; Eslami et al., 2018; Singh et al., 2019; Yoon et al., 2020). Some studies have focused on using spatial information to improve navigation performance (Zhang et al., 2017; Rosenbaum et al., 2018), while others have aimed to construct cognitive maps (Eslami et al., 2018; Singh et al., 2019; Yoon et al., 2020). In this study, we investigate incorporating spatial information into episodic memory. We show that this can improve performance and memory usage efficiency on spatial reasoning tasks without sacrificing performance.

## 5 Conclusion and Limitations

Transformer-based episodic memory has utilized temporal order to serialize experience frames. In this paper, we explore the potential of incorporating the spatial axis as another fundamental aspect of the physical world. We argue that this is crucial for embodied agents and introduce the concept of Spatially-Aware Transformers (SAT). We propose different SAT architectures, starting from the simplest SAT-FIFO to SAT with place memory, which includes a hierarchical episodic memory centered around places. Furthermore, we introduce the Adaptive Memory Allocation (AMA) method, which provides a more flexible memory management strategy beyond the FIFO memory writing approach. Through experiments, we assess the improved performance of these methods and demonstrate their applicability to a wide range of machine learning problems.

**Limitations.** Despite the aforementioned contributions, this work is based on several simplifying assumptions as the first to propose the concept of spatially-aware transformers. One of the main assumptions is its reliance on spatial annotation. Although spatial annotation is widely available in many embodied agent applications, incorporating the learning of spatial representation alongside its usage could enhance the applicability of the proposed models. Another limitation is the generality of the AMA method. While it offers more flexibility than FIFO, it still relies on a set of predefined strategies. While it would be ideal to discover the strategy from scratch, previous works like the Neural Turing Machine (Graves et al., 2014; 2016) have demonstrated that this approach can introduce significant training complexity, limiting its practical application. Thus, there is a need for a new method that improves both flexibility and ease of training. Lastly, while in this work we focused on spatial reasoning tasks, there would also be non-spatial reasoning tasks in spatial environments. Generalizing the proposed method to handle such cases would also be an interesting future direction.

## 6 ETHICS STATEMENT

In this paper, we present a fundamental and general backbone for embodied agents using a transformer that incorporates spatial information. Given the nature of this work, it does not pose any immediate ethical risks as it is primarily a theoretical contribution to the field of machine learning. However, we acknowledge the potential for this work to be applied in the development of autonomous agents in the future. While such applications could have numerous benefits, they could also raise ethical concerns depending on the context and manner of use. We strongly advocate for the responsible and ethical use of this and any other AI technology.

## 7 REPRODUCIBILITY STATEMENT

To facilitate the reproducibility of our work, we have provided the pseudo codes, model architecture and hyperparameters in Appendix B.3, and B.4. We will also release the source code for our models and experiments.

## ACKNOWLEDGEMENT

This work is supported by Brain Pool Plus (BP+) Program (No. 2021H1D3A2A03103645) through the National Research Foundation of Korea (NRF) funded by the Ministry of Science and ICT.

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

## A   COMPUTATIONAL OVERHEAD

Our study was performed on an Intel server equipped with 8 NVIDIA RTX 3090 GPUs and 256GB of memory. The training duration for the tasks in the Room Ballet environment was approximately 12 hours on a single GPU. The generation and reinforcement learning tasks required approximately 3 days and 12 hours on a single GPU, respectively.

## B   MODEL DETAILS

### B.1   TRANSFORMER DESIGN

We utilized a memory-equipped transformer architecture, similar to the Transformer-XL (TrXL) model (Dai et al., 2019). In this architecture, the query, key, and value are not identical. Instead, the key and value are maintained as the equipped memory which is the same at every layer and is represented by the sum embedding of inputs such as the observation or place information. The query is a separate input, which is usually a single observation. This architecture avoids the quadratic computation required by naive transformers, but it does require attending to every memory, which can be a burden in reinforcement learning tasks (Parisotto & Salakhutdinov, 2021).

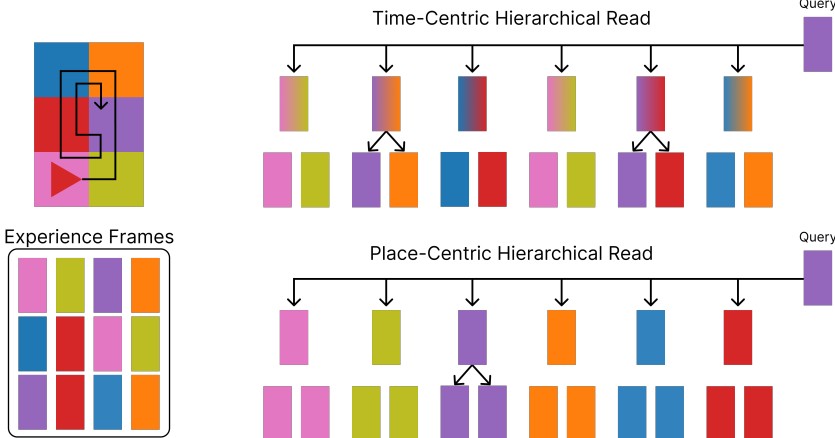

**Figure 9:** The agent randomly explores the places represented by different colors, as shown in the top left figure. The collected experience frames are shown in the bottom left figure. When retrieving the memory using the time-centric hierarchical read, the chunks contain memories from multiple places, and memories from the same place are distributed across multiple chunks. This makes it difficult to retrieve place-centric knowledge from the chunks. In contrast, the place-centric read operation ensures that each chunk contains memories from the same place, making it easier to retrieve place-centric knowledge than with the time-centric read operation.

### B.2   HIERARCHICAL READ OPERATION

Our hierarchical read operation is inspired by the Hierarchical Chunk Attention Memory (HCAM) Lampinen et al. (2021). The memory is structured into chunks, which are sets of experience frames. The chunk size is a hyperparameter. In HCAM, each chunk consists of temporally contiguous memories, such as the memories from time step t to t+n. In our place-centric hierarchical read operation, each chunk is a set of temporally contiguous memories that are associated with the same place. For example, if the agent visits a place at time steps t, t+2, and t+4, then those memories will form a chunk, even though the memories at time steps t+1 and t+3 are not included. Each chunk has its own representative vector, which is the average of the memories in the chunk.

The read operation is the same as that in HCAM. At each layer, the query is attended to the chunk representatives, and the top-k most similar chunks are selected. The query is then attended to the memories in the top-k chunks, and the output of the read operation is represented by a weighted average of the chunk representatives and the memories in the chunks, where the weights are determined by the similarities. By selecting the top-k chunks, the hierarchical read operation can

reduce the computational overhead of reading the memory. More detailed descriptions on hierarchical attention mechanism is given as follows.

**Hierarchical Attention Mechanism** Now, we explain how query extracts the relevant information from place memory. SAT with place-centric hierarchical read operation constructs a spatially-aware memory $M = \{M^i\}_{i=1}^N$, where $M^i$ is a place-wise episodic memory per *place* $i = 1, \ldots, M$. Each $M^i$ stores $N_i$ chunks, and we denote $j$-th chunk in $M^i$ as $C_{ij}$. Specifically, At time step $t$, when experience frame $x_t$ in *place* $i$ is given, we store the experience frame in the most recent chunk (i.e., $C_{iN_i}$). If the chunk is full, then new empty chunk is allocated in $M^i$.

By constructing place memory $M$ from past observations, when normalized query $q$ is given, it can use chunk representatives to choose top-$k$ relevant chunks, and then attending within those most relevant chunks in further detail. In particular, let $C_{ij}$, $e_i^{place}$ denote the $j$-th chunk in $M^i$, and a place embedding at place $i$, respectively. Then the chunk relevance ($R_{ij}$) is computed as:

$$R_{ij} = \text{softmax}(\text{Linear}(q) \cdot \text{Linear}(K)), \quad K = \text{Combine}(\mu(\mathcal{C}_{ij}), e_i^{place}) \tag{1}$$

where Linear$(\cdot)$ is a linear layer, $\mu(\cdot)$ is average over experience frames in chunk. Combine$(\cdot, \cdot)$ can be any operation such as concatenation and addition, while we used addition in actual implementations. After retrieving chunks with top-$k$ relevance scores, the model attends the detail within the chunks. Then, output query result $q^*$ is computed as:

$$q^* = \sum_{i,j \in \text{top-}k \text{ from } R_{ij}} R_{ij} \cdot \text{MHA}(q, \mathcal{C}_{ij}, \mathcal{C}_{ij}) \tag{2}$$

where MHA$(\cdot, \cdot, \cdot)$ is multi-head attention that takes query, key, value inputs in order. This attention mechanism lets query to select the most relevant memory chunks through space and time. For instance, query can extract information such as *what happened in the left room of given context?*. Note that when the number of distinct place $M$ is 1, the formulation reduces to HCAM.

### B.3 PSEUDO CODES

We introduce the pseudo-codes of SAT-AMA, SAT(-FIFO), DNC training for a supervised learning task in Algorithm. 1-3.

---

**Algorithm 1** Spatially-Aware Transformer with Adaptive Memory Allocator

---

**Require:** $\mathcal{T}$: a set of tasks with the task descriptions $\tau$. $\{o_t^\tau, l_t^\tau\}_{t=1}^L$: a series of observations and location annotations. $(q, y)$: a query image and the true label. $\mathcal{L}(.,.)$: the downstream task loss. $\mathcal{A}$: a set of memory allocation strategies. $\epsilon$: a hyperparameter to explore. $\theta$: the parameters for SAT and a classifier. $f_\theta(.)$: a classifier for the supervised learing task. $\phi$: the parameters for AMA policy.

1: Initialize the parameters
2: **while** until converge **do**
3:     Sample a mini-batch of tasks $\mathcal{T}_b$ from $\mathcal{T}$
4:     **for** each task $\tau \in \mathcal{T}_b$ **do**
5:         With probability $\epsilon$ select a random rule $\sigma^\tau$, otherwise select $\sigma^\tau = \arg\max_{\sigma \in \mathcal{A}} Q_\phi(\tau, \sigma)$
6:         Initialize $\mathcal{M}_0^\tau$ as empty.
7:         **for** $t = 1, \ldots, L$ **do**
8:             $x_t^\tau = \texttt{sum\_embed}_x(\texttt{embed}_l(l_t^\tau), \texttt{embed}_t(t), \texttt{embed}_o(o_t^\tau))$
9:             Memorize $\mathcal{M}_t^\tau = \texttt{Update}(\mathcal{M}_{t-1}^\tau, x_t^\tau, \sigma^\tau)$
10:         **end for**
11:     **end for**
12:     $\phi \leftarrow \phi - \beta \nabla_\phi \sum_{\tau \in \mathcal{T}_b} (Q_\phi(\tau, \sigma) + \mathcal{L}(f_\theta(\texttt{Read}(\mathcal{M}_L^\tau, q^\tau)), y^\tau))^2$
13:     $\theta \leftarrow \theta - \alpha \nabla_\theta \sum_{\tau \in \mathcal{T}_b} \mathcal{L}(f_\theta(\texttt{Read}(\mathcal{M}_L^\tau, q^\tau)), y^\tau)$
14: **end while**

---

### B.4 MODEL ARCHITECTURES AND HYPERPARAMETERS

**Model Architecture** Our model architecture is based on the HCAM model Lampinen et al. (2021). HCAM model is a stack of memory layers, and each memory layer consists of:

---

**Algorithm 2** Spatially-Aware Transformer without AMA

---

**Require:** $\mathcal{T}$: a set of tasks with the task descriptions $\tau$. $\{o_t^\tau, l_t^\tau\}_{t=1}^L$: a series of observations and location annotations. $(q, y)$: a query image and the true label. $\mathcal{L}(., .)$: the downstream task loss. $\theta$: the parameters for SAT and a classifier. $f_\theta(.)$: a classifier for the supervised learing task.

1:  Initialize the parameters
2:  **while** until converge **do**
3:      Sample a mini-batch of tasks $\mathcal{T}_b$ from $\mathcal{T}$
4:      **for** each task $\tau \in \mathcal{T}_b$ **do**
5:          Select some strategy $\sigma^\tau$ (e.g., FIFO)
6:          Initialize $\mathcal{M}_0^\tau$ as empty.
7:          **for** $t = 1, \ldots, L$ **do**
8:              $x_t^\tau = \texttt{sum\_embed}_x(\texttt{embed}_l(l_t^\tau), \texttt{embed}_t(t), \texttt{embed}_o(o_t^\tau))$
9:              Memorize $\mathcal{M}_t^\tau = \texttt{Update}(\mathcal{M}_{t-1}^\tau, x_t^\tau, \sigma^\tau)$
10:         **end for**
11:     **end for**
12:     $\theta \leftarrow \theta - \alpha \nabla_\theta \sum_{\tau \in \mathcal{T}_b} \mathcal{L}(f_\theta(\texttt{Read}(\mathcal{M}_L^\tau, q^\tau)), y^\tau)$
13: **end while**

---

**Algorithm 3** DNC

---

**Require:** $\mathcal{T}$: a set of tasks with the task descriptions $\tau$. $\{o_t^\tau, l_t^\tau\}_{t=1}^L$: a series of observations and location annotations. $(q, y)$: a query image and the true label. $\mathcal{L}(., .)$: the downstream task loss. $\theta$: the parameters for DNC and a classifier. $f_\theta(.)$: a classifier for the supervised learing task.

1:  Initialize the parameters
2:  **while** until converge **do**
3:      Sample a mini-batch of tasks $\mathcal{T}_b$ from $\mathcal{T}$
4:      **for** each task $\tau \in \mathcal{T}_b$ **do**
5:          Initialize memory matrix $M_0^\tau$ as empty.
6:          **for** $t = 1, \ldots, L$ **do**
7:              $x_t^\tau = \texttt{cat\_embed}_x(\texttt{embed}_l(l_t^\tau), \texttt{embed}_o(o_t^\tau))$
8:              Update $M_t^\tau = \texttt{Write}(M_{t-1}^\tau, x_t^\tau)$
9:          **end for**
10:     **end for**
11:     $\theta \leftarrow \theta - \alpha \nabla_\theta \sum_{\tau \in \mathcal{T}_b} \mathcal{L}(f_\theta(\texttt{Read}(M_L^\tau, q^\tau)), y^\tau)$
12: **end while**

---

1. A local attention (LA) block, which uses self-attention within query vectors as in the transformer Vaswani et al. (2017).

2. A hierarchical chunk attention memory (HCAM) block, which allows the output query vectors from the LA block to perform cross-attention on hierarchical chunk memory.

3. A multilayer perceptron (MLP) block, which processes the final query output.

For each block, we use a residual connection of the input, and layer normalization is applied before the block Parisotto et al. (2020). The transformer and SAT architectures are implemented by selecting the entire chunk at a high level with a chunk size of 1. Different from HCAM, we use the same memory at each layer, which is represented by a sum embedding as discussed in Appendix B.1. Additionally, while HCAM stops the gradients through the memory, our model can be learned from the gradients through the memory.

**Common Hyperparameters** For all supervised learning tasks, we used a batch size of 32, Adam optimizer Kingma & Ba (2014) with a learning rate of 0.0002, and gradient clipping over the range $[-5.0, 5.0]$. We stacked four memory layers, each of which consists of an LA block, an HCAM block, and an MLP block. We used a dimension of 128 for embedding vectors. For the LA and HCAM blocks, we used 2 heads and a head dimension of 64 for multi-head attention operations. For the MLP block, we used a 2-layer MLP with a hidden dimension of 128, and a ReLU Nair & Hinton (2010) activation function. For the Q network, we used a 1-layer MLP with a hidden dimension of 64

and a ReLU activation function. We also used $\epsilon$-annealing for Q-learning Mnih et al. (2015), starting from $1.0$ and decreasing to $0.2$ over $200,000$ steps.

## C  TASK DETAILS

### C.1  INPUT AND EXPECTED OUTPUT FOR THE TASKS

As delineated in the main text, the agent acquires a sequence of inputs - observation, time-step and space information - as the agent moves randomly except **Exp-6**. The agent movement in **Exp-6** is discussed in Section C.6. Following these inputs, a query is presented. For the Room Ballet and MiniWorld Environment, the query is an image showing a dancer's appearance. For image observations and query image, we used a CNN for the encoding. Then we added time and spatial embedding on encoded observations and feeding encoded observations and query to transformer as illustrated in Figure 9. For the action-conditioned generation task, (**Exp-5**), the query consists of the observation at the starting position, and a sequence of actions for multi-step generation. Here, we used a learnable embedding for actions, and remainings are same. In reinforcement learning (RL) tasks, the query is the observation encountered after returning to the starting room. The encoding process is same as in Room Ballet environment.

For the tasks where the memory capacity is limited such as **Exp-3** and **Exp-4**, additional task information is imparted at the outset of the episodes, either in numerical form or as narrative descriptions. The agent must employ its cognitive faculties to process and integrate these various inputs within the given constraints, such as limited memory capacity.

The agent's outputs are tailored to the nature of the task at hand: for the Room Ballet environment, it predicts the *dance type*; for image generation tasks, it produces a sequence of images; and for RL tasks, it infers an action at each time-step.

### C.2  PLACE EMBEDDING FOR THE TASKS

**Room Ballet (Exp-1-4)** The room formation is $3 \times 3$ to $5 \times 5$, and we used a 1D sinusoidal embedding for place embedding. If room location is $(3, 2)$, index $8$ in sinusoidal embedding is used.

**FFHQ Environment (Exp-5)** The grid size is $10 \times 10$. Here, we exploit 2D structure in place embedding. In particular, we posed a disjoint sinusoidal positional embeddings for two sets, which are $\{1, 2, 3, 4, 5, 6, 7, 8, 9, 10\}$, and $\{11, 12, 13, 14, 15, 16, 17, 18, 19, 20\}$ for $x$ and $y$ axis, respectively. For example, if the room position is $(x, y) = (2, 3)$, we make the embedding as the sum of sinusoidal positional embedding at $2$ and $13$. More detailed discussion is in Appendix D.9.

**MiniWorld Environment (Exp-6)** We divided the map into $4$ places, where each place covers 90 degrees range of agent's view (The agent can only rotate here). We used a 1D sinusoidal embedding.

**RL Agent (Sec. 3.3)** The environment consists of two rooms, and we used a 1D sinusoidal embedding.

### C.3  BALLET TASK WITH AN ACTION AT EACH TIME-STEP (SHORT-STAY TASK)

The agent can randomly move one of four actions, up/down/left/right at each time-step in this task. When it's movement is blocked by the wall, then it does not move. We tested $3 \times 3$, $4 \times 4$, and $5 \times 5$ rooms settings. The maximum movements are 1280, which is the same as the total observations in the Next Ballet task in Exp-1. The chunk size is 32, which is the same as the length of the dance to fit the configuration as same as other settings in Exp-1, 3 and 4.

### C.4  BALLET TASKS IN EXP-3

As illustrated in Figure 10, in Ballet-FIFO/LIFO, the agent is asked to predict the dance type that was observed most recently or in the past, respectively. In Ballet-MVFO/LVFO, the agent is asked to predict the most recently observed dances within each room and the dances in the most frequently visited room, respectively.

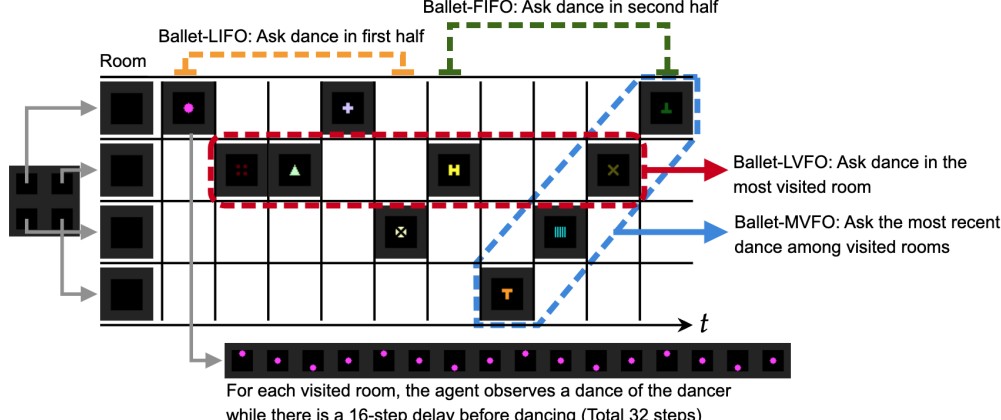

**Figure 10:** The visualization of Ballet tasks in Exp-3 for 4-room case.

## C.5    TASK DESCRIPTIONS FOR SAT-AMA GENERALIZATION EXPERIMENT

We utilized 20 task descriptions for each task ID for **Q4**. The task descriptions are in Tables 4 and 5.

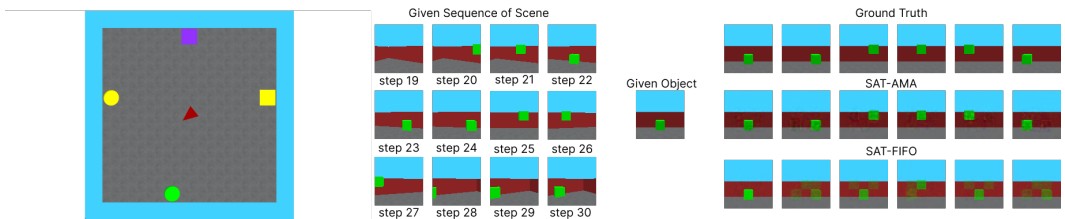

**Figure 11: Left:** The overview of the room. **Right:** The generated samples from SAT-FIFO and SAT-AMA. While SAT-FIFO failed to generate the dance correctly, SAT-AMA generated well even though it saw the dancing while rotating.

## C.6    GENERATION ON MINIWORLD

In this task, the aim is to generate a dance sequence based on observations of objects on North, West, East and South sides in the MiniWorld environment (Chevalier-Boisvert, 2018). The agent begins by turning around from the North West side at a random degree between 3 and 7, and stops to observe three objects dancing on the South side. The agent then returns to the initial angle and is asked to generate the dance sequence that was observed. The total number of steps is 126, with the agent turning 45 steps first, watching the dancing for 54 steps (during which three objects dance per 18 steps), and then turning again for 27 steps. The objects are chosen randomly from two shapes, ball and box, and six colors, including red, green, blue, purple, yellow, and grey. The dance sequences are chosen randomly from five options: circle clockwise (move left - move up - move right - move right - move down - move left), circle counterclockwise (move right - move up - move left - move left - move down - move right), move up and down (move up - move down - move up - move down - move up - move down), move left and right (move left - move right - move right - move left - move left - move right), and move top-left and top-right (move top-left - move bottom-right - move top-right - move bottom-left, move top-left, move bottom-right). The length of each dance sequence is 6.

The given task is one of FIFO/LIFO/MVFO/LVFO tasks and use a common architecture in Appendix B.4, except for the encoder and decoder. The encoder consists of three convolutional layers with hidden sizes of [64, 64, 16], kernel sizes of [8, 4, 3], and strides of [4, 2, 3]. The activation function used is LeakyReLU (Maas et al., 2013) with a slope of 0.01. After the convolutional layers, a single linear layer with a hidden size of 128 is used. The decoder architecture consists of one linear layer with a hidden size of 144 and ReLU activation, followed by four transposed convolutional layers with hidden sizes of [64, 64, 18], kernel sizes of [3, 4, 8], strides of [3, 2, 4], and two ReLU activations, as

well as a single Tanh activation. It should be noted that the last layer's hidden size is derived from the image channel size multiplied by the length of the dance sequence, which is 6 in this case.

## D  ADDITIONAL EXPERIMENTAL RESULTS

|  | Inference Time (ms) |
|---|---|
| SAT | $8.43 \pm 1.37$ |
| SAT-FIFO-TH | $7.51 \pm 1.58$ |
| SAT-PM-PH | $7.49 \pm 1.29$ |

**Table 1:** The inference time comparison between SAT, SAT-FIFO-TH, and SAT-PM-PH. It is the inference time only for SAT `Read` and the average value for 1000 samples. SAT requires to attend the every memory, it requires much more time to do inference.

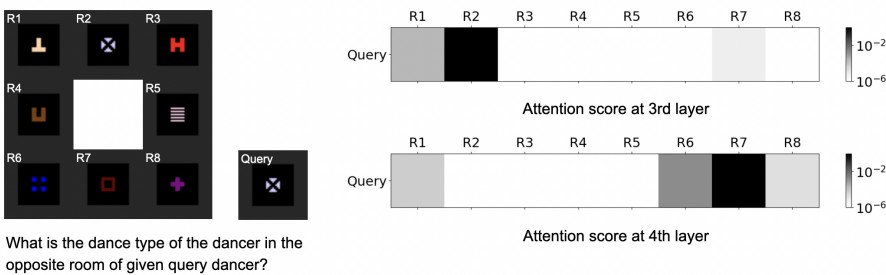

**Figure 12: Left:** The illustration of the Opposite Ballet task. **Right:** The attention scores at 3rd and 4th layers.

### D.1  ATTENTION MAP VISUALIZATION IN SAT

To verify that SAT can address the spatial reasoning task more clearly, we investigated the attention weight map visualization in SAT when solving a spatial reasoning task. To do this, we introduced a new task in the Room Ballet environment called the Opposite Ballet task. This task is similar to the Next Ballet task in Exp-1, but there is no dancer at the center and the agent cannot go there. The agent observes every dance in each room, and is asked to predict the dance that the dancer did *in the opposite room* of the given dancer, as depicted in the left figure of Figure 12.

We captured the attention weight map from SAT as shown in the right figure of Figure 12. We observed that the query attends to R2 (query dancer's room) in the 3rd layer (to extract the space knowledge in this attention), and then attends to the opposite room (R7) in the 4th layer. We note that the query has no spatial information. This result implies that to do spatial reasoning, SAT requires several layers. If SAT is implemented with multiple layers, it can infer the spatial reasoning even when the spatial information is not given in the query.

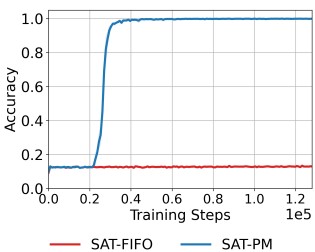

**Figure 13:** Accuracy on Ballet-ABA task. It shows that SAT-PM can handle the scenario (b) in Figure 1 while SAT-FIFO fails.

### D.2 EMPIRICAL EVALUATION OF THE SCENARIO (B) IN THE FIGURE 1 WITH SAT-PM

We discussed the Place Memory can handle the long-stay issue in the scenario (b) in the Figure 1 in the Section 2.2. Here, we empirically evaluated it on the Room Ballet environment. To do this, we created a new task, Ballet-ABA task. In this task, the agent visits a total of 18 rooms (576 steps), with the capacity to store observations from only 8 rooms (256 steps). The agent initially visits 7 unique rooms, and subsequently remains in the last room for the duration of the task. The agent is then asked to predict the dance type of the given query dancer image in the initial 7 rooms.

In this scenario, SAT-FIFO struggles to retain the necessary information for accurate predictions. In contrast, SAT-PM demonstrates superior performance, as shown in Figure 13. This underscores the advantage of utilizing spatially-aware memory storage operations when faced with memory limitations in spatial tasks.

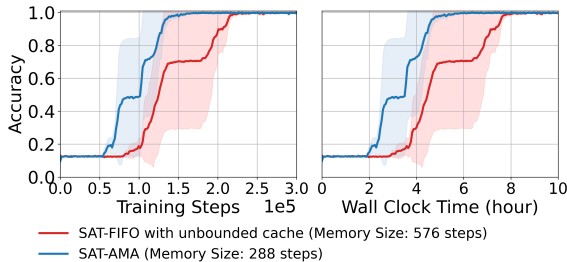

**Figure 14:** Performance comparison between SAT-FIFO with unlimited memory and SAT-AMA on Ballet Strategy tasks. In this task, one of four tasks (Ballet-FIFO/LIFO/MVFO/LVFO) is given for each episode as in Exp-3. SAT-AMA has to learn to choose the proper memory allocation strategy for the given task ID, while SAT-FIFO with unlimited memory requires retrieving larger memory.

### D.3 PERFORMANCE COMPARISON BETWEEN SAT-AMA AND SAT-FIFO WITH UNLIMITED MEMORY

In Exp-3, we modified the problem setting to limited memory to show efficient memory usage of AMA. This made SAT-FIFO with limited memory to deteriorate in Ballet-Strategy tasks. One may ask that if SAT-FIFO with unlimited memory can still solve the problem, however, we show that SAT-AMA outperforms SAT-FIFO in speed even SAT-FIFO has unlimited memory capacity.

To verify this, we evaluated SAT-AMA and SAT-FIFO with unlimited memory on Ballet Strategy tasks in Exp-3. In Figure 14, we can see that, even if it requires additional learning for the AMA policy, SAET-AMA can learn the task faster because it maintains a small amount of memory. This result shows that learning through comparison with more memory slots is less efficient than learning with limited memory.

| Strategy | FIFO | LIFO | MVFO | LVFO |
|---|---|---|---|---|
| AMA Distribution | 0.475 | 0.073 | 0.150 | 0.302 |
| Accuracy | 0.999 | 0.146 | 0.325 | 0.714 |
| Normalized Accuracy | 0.457 | 0.067 | 0.149 | 0.327 |

**Table 2:** The relationship between AMA strategy distribution and task performance for the Ballet-FIFO task.

### D.4 THE CORRELATION BETWEEN STRATEGY CHOICES AND TASK PERFORMANCES

To more deeply understand how Adaptive Memory Allocator (AMA) works, in this section, we illustrate how AMA adapts its strategy selection to different experimental settings. To address this, we investigate further to show the distribution of strategies chosen by AMA, and discuss its dependency on the nature of tasks, and the correlation between strategy choices and task performance. The table 2 presents the distribution of strategies selected by AMA, along with corresponding performance metrics, derived from the Ballet-FIFO task in **Exp-3**. We note that each column represents the

probability of each strategy being selected, the accuracy when each strategy was selected. The last row represents the accuracies normalized for the four strategies.

The strategy distribution was averaged across three distinct runs (random seeds) and was ascertained by passing the output of the AMA's policy network through SoftMax. The results elucidate a clear correlation between the chosen strategies and task performance, reinforcing the premise that AMA's strategic choices are inherently linked to efficacy.

## D.5 GENERALIZATION PERFORMANCE FOR THE UNSEEN SIZE OF ENVIRONMENT

To ascertain the model's generalization capabilities, we assessed performance variations under conditions where the environment size or memory capacity differed from the training phase. Our SAT model employs hierarchical memory organized in chunks (where each chunk is a set of observations) and is designed to retrieve the most pertinent top-K chunk memories in response to a query. Consequently, the model is adaptable to changes in memory size, reflected by the number of chunks, ensuring consistent application.

In the Ballet-MVFO task, detailed in **Exp-3**, the agent is trained in an environment with 8 rooms, visiting each room 18 times to observe the dancers' performances (remaining in each room for 32 steps and viewing the dance for half of the duration). The agent's memory is sized to accommodate observations from 9 rooms, enabling it to recall the top-$K$ (K=4) relevant chunks—equivalent to observations from 4 rooms—based on the query provided, to infer the dance sequence.

To validate the model's adaptability to both environmental and memory size changes, we conducted supplementary experiments by varying the number of rooms to 4, 6, 10, and 12, using three random seeds for each scenario, and adjusted the memory size to match the number of rooms.

As Table 3 illustrates, the model demonstrated a marginal increase in accuracy in environments with fewer rooms than the training setup, indicating an effective generalization. Moreover, the performance remained robust even with an increased number of rooms, showcasing the model's resilience to environmental scale variations.

|  | 8 Rooms (Train Env.) | 4 Rooms | 6 Rooms | 10 Rooms | 12 Rooms |
|---|---|---|---|---|---|
| Accuracy | 99.54% | 99.86% | 99.76% | 99.19% | 98.04% |

**Table 3:** Generalization performance of our model in environments of different sizes compared to the training environment.

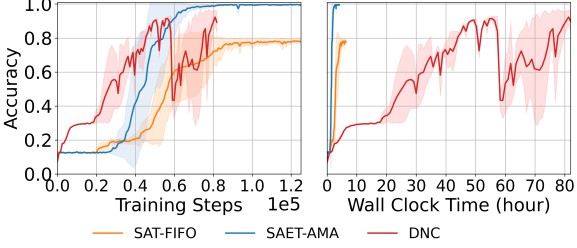

SAT-FIFO    SAET-AMA    DNC

**Figure 15:** Performance comparison between DNC and SAT-AMA/FIFO on Ballet-MVFO task. SAT-AMA, while DNC converges faster than SAT-AMA, the performance is worse, and it is much slower than SAT-AMA.

## D.6 PERFORMANCE COMPARISON WITH DIFFERENTIABLE NEURAL COMPUTER (DNC)

One of the limitations of our work is that we learn AMA policy from pre-defined set of strategies. In ideal case, the model should discover the strategy from scratch with learning-based read/write mechanisms such as Neural Turing Machine (NTM) (Graves et al., 2014). However, it is known that learning-based read/write memory is slow and difficult to optimize in practice.

To investigate the learning-based read/write memory, we applied Differentiable Neural Computer (DNC) (Graves et al., 2016) to some of our tasks, which is an upgraded version of NTM. For this

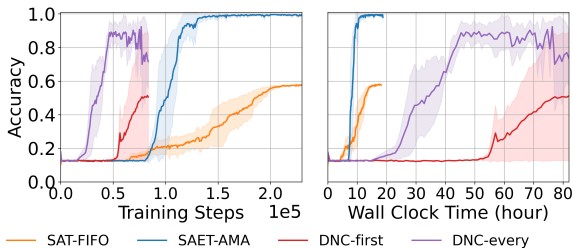

**Figure 16:** Performance comparison between DNC and SAT-AMA/FIFO on Ballet-MultiTask in **Exp-3**. DNC-first takes task ID as learnable embedding only at first time step, while DNC-every takes task ID as one-hot vector concatenated to an encoding of the observed image for every time step. While DNC-every converges faster than SAT-AMA, the performance is worse, and has much lower speed than SAT-AMA. DNC-first which has the same task setting as SAT-AMA (i.e., giving the task information only at initial step) performed worst.

investigation, it is evaluated for Ballet-MVFO and Ballet-MultiTask in **Exp-3**. To apply this to DNC, The observation encoding is same to our model. Originally, DNC is not designed for the multi-tasks setting, we added the task information in the two ways. One of them, DNC-first is giving the task information at the beginning of the episode as done for our model. Another, DNC-every is concatenating the task information to the observation embedding at every timestep. It is designed to prevent that the DNC forgets the task information because the backbone of DNC is the recurrent module which is limited to handle the long-term dependency. The DNC's memory size is same to SAT-AMA, 288 cells with 128 dimensions each. LSTM was used for DNC's controller, and the sizes of image representation and representation in the hidden layer were set to 128.

The results are shown in Figures 15 and 16. Commonly, DNC showed more efficient training in the aspect of the training steps, while it is tremendously slower in the wall-clock time comparison. It is because the backbone of DNC is consisted of the recurrent module, which training is slower than the training of Transformer that is the backbone of SAT-AMA. The reason why SAT-AMA is slower in the training steps is that the policy of AMA should requires some exploration at the early stage of the training. Additionally, DNC-every outperforms DNC-first, and it shows sub-optimal performance. This result suggests that DNC is limited to handle the long-term dependent knowledge. While the result on DNC models may require more training steps for better analysis, the result shows the limitations of the current learning-based read/write memory module. However, as stated in Limitations, investgating more advanced forms of learnable memory as a strategy allocator would be an interesting future work.

### D.7    SAT-PM WITH APPROXIMATE PLACE CLUSTERS IN FFHQ ENVIRONMENT

We investigate the effect of the approximate place structure on SAT-PM in FFHQ environment. In Room Ballet tasks, a place was already clustered in rooms, and the agent could easily manage the place memory by storing the observations in each room separately. However, in FFHQ environment, there is no pre-defined place structures. In realistic scenario, the agent should construct its own place clusters based on given locations with the clustering algorithm (e.g., k-nearest neighbor), then use the place structure for the downstream tasks.

To show that SAT-PM is robust to approximate place clusters, we designed three types of place clusters in FFHQ environment as depicted in Figure 17 (b). To generate place clusters, we applied a k-nearest neighbor algorithm with different number of clusters ($N = 8, 16, 32$) on a set of locations that is available in the environment (i.e., 100 locations in $10 \times 10$ grid map). Depending on the number of place clusters, we denote the SAT-PM model using those place clusters as SAT-PM($N$). As described in Exp-6, SAT-PM(8), SAT-PM(16), and SAT-PM(32) showed comparable performance as SAT-FIFO. This implies that we do not need the ground truth place structure but an approximate structure would be enough in practice. In Fig. 17 (c), we present the full results of generated samples. We note that SAT-PM with approximate place clusters perform as well as SAT-FIFO.

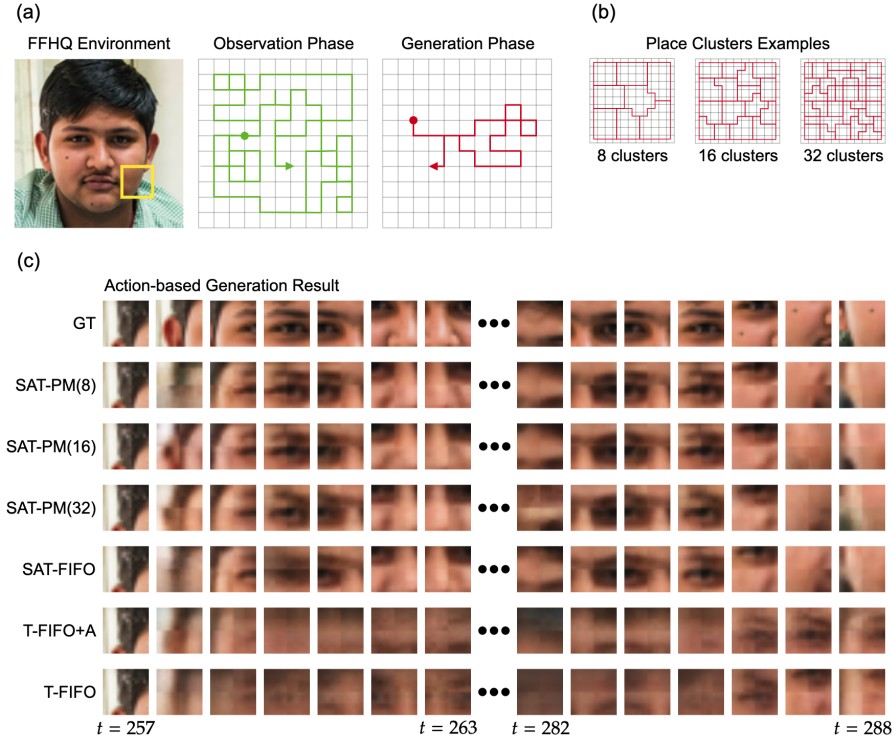

**Figure 17:** (a) FFHQ environment. The agent freely navigates in $10 \times 10$ grid map observing partial scene (yellow box). In Observation phase, the agent randomly navigates for 256 steps (only the first half is visualized in green trajectory). In Generation phase, the agent should generate the scenes given starting location's scene and action sequence (red trajectory). Circled locations are agent's initial location in each phase. (b) Place clusters examples. We use three types of place clusters with different number of clusters using knn algorithm on a set of available locations in grid map. (c) The generation result of all models.

## D.8 SAT-AMA WITH APPROXIMATE PLACE CLUSTERS IN FFHQ ENVIRONMENT

As Exp-3 in Room Ballet environment, we can extend the FFHQ environment to limited memory setting where the choosing the proper strategy based on downstream task becomes crucial. To do this, we double the length of Observation phase to 512 steps, and assume that only half of total steps (i.e., 256 steps) can be stored in memory. In Observation phase, the agent stays in $2 \times 2$ region for 256 steps in the middle of random walk. This scenario is from thought experiment in Figure 1 (b), where FIFO memory would fail to store full information of the facial image. However, AMA policy can choose place-centric memory which enables to store full information of the image.

For evaluation, we compared SAT-FIFO and SAT-AMA. When place-centric strategy (MVFO) is chosen by SAT-AMA, we used approximate place clusters ($N = 8$) in Sec. D.7. In Figure 18 (a), we note that SAT-AMA with approximate place clusters outperform SAT-FIFO by large margin. In Figure 18 (b), the visualization of generation results show that SAT-AMA could generate the scenes well, while SAT-FIFO fails to. This implies that SAT-AMA with approximate place clusters still work in action-conditioned generation task.

## D.9 COMPARATIVE ANALYSIS OF EMBEDDING TYPES FOR SPACE EMBEDDING FOR THE GENERATION TASK ON FFHQ ENVIRONMENT

In this paper, we utilized the sinusoidal positional embedding (Vaswani et al., 2017) for simplicity. However, the space is not uni-directional. For example, it can be represented as x and y coordinates. Therefore, we investigated 2 dimensional positional embedding in this section.

We implemented two 2-D positional embedding; 2-D sinusoidal and learnable positional embedding. The 2-D sinusoidal positional embedding is constructed by summing sinusoidal embeddings corre-

(a)

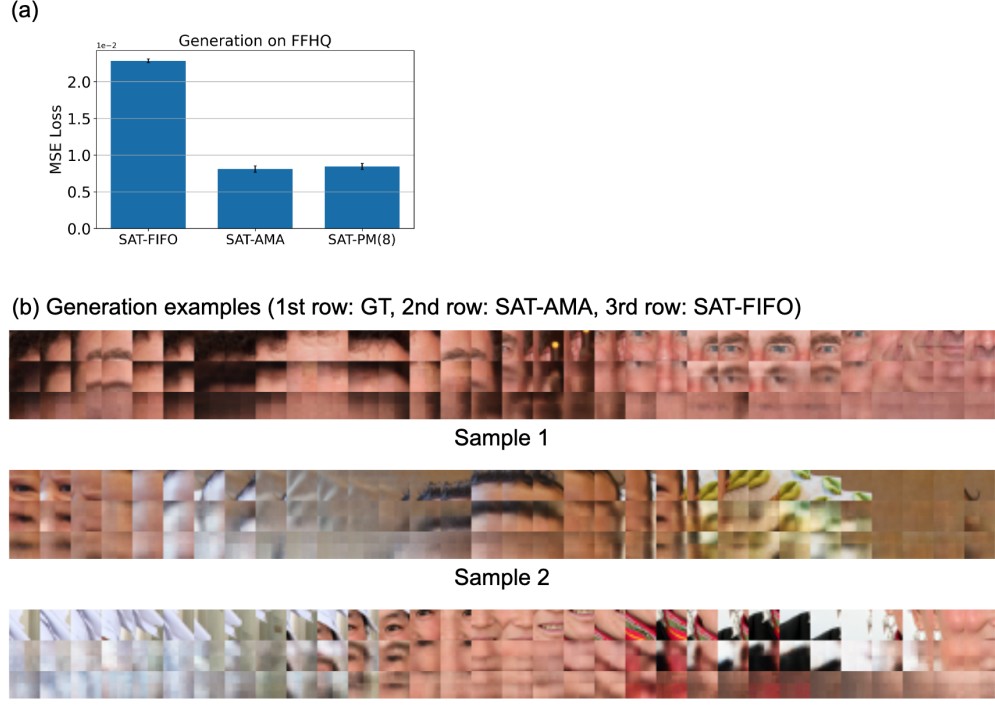

(b) Generation examples (1st row: GT, 2nd row: SAT-AMA, 3rd row: SAT-FIFO)

Sample 1

Sample 2

Sample 3

**Figure 18:** (a) The generation performance on FFHQ environment when memory is limited. SAT-AMA outperforms SAT-FIFO by large margin. (b) Generation examples of SAT-AMA, and SAT-FIFO.

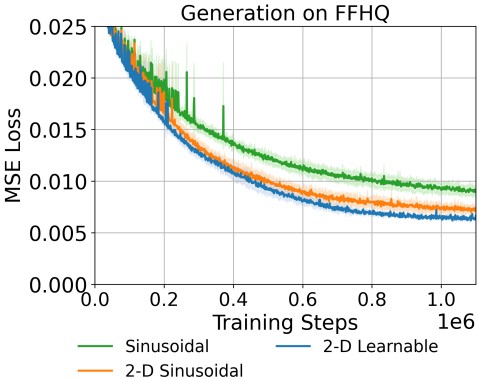

**Figure 19:** MSE loss comparison of sinusoidal, 2-D sinusoidal, and 2-D learnable embeddings for spatial embedding. The task pertains to image generation in the FFHQ environment, as detailed in Section 3.2. The figure demonstrates the superior performance of the 2-D learnable embedding over the 2-D sinusoidal, with both significantly outperforming the standard sinusoidal embedding. This improvement is attributed to the non-unidirectional nature of spatial dimensions.

sponding to each axis. For instance, at position (x,y) = (0,1), we combine the 0th index sinusoidal embedding for the x-axis with the 1st index embedding for the y-axis. Notably, this method produces non-symmetric embeddings for symmetric positions, such as (0,1) and (1,0), due to the addition of a large number to the index at the y-axis to ensure differentiation.

The 2-D learnable embedding follows a similar structure but is optimized through backpropagation based on the downstream task's loss. It is designed with distinct embeddings for the x and y axes, enabling discrimination of symmetric positions.

As illustrated in Figure 19, our results indicate a clear advantage of 2-D embeddings over the sinusoidal positional embedding (Vaswani et al., 2017), with the learnable 2-D variant exhibiting marginally better performance than the sinusoidal 2-D. These findings suggest that a spatial embedding approach that more accurately reflects the bidirectional nature of space is preferable. Interestingly, the additional flexibility afforded by learnability does not yield a significant benefit, implying that the inherent structure of 2-D embeddings may be sufficient to capture the spatial relationships pertinent to the task.

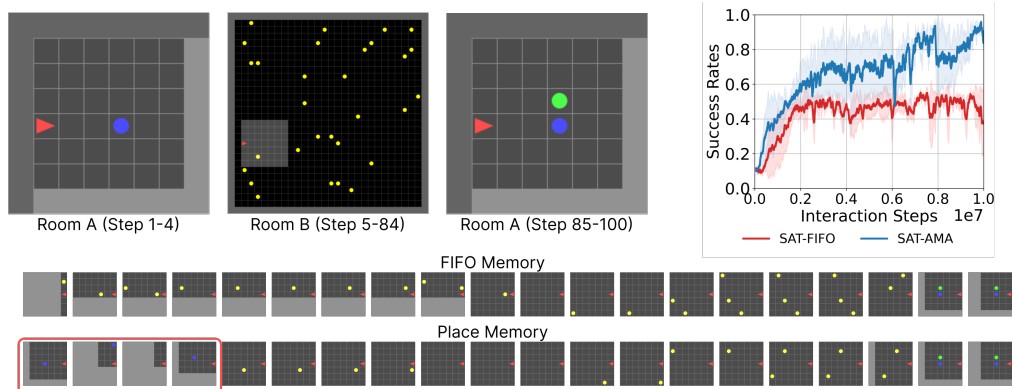

**Figure 20: Top Left:** The Visual Match Task Visualization. In room A, the agent can see the goal ball color, and after lots of steps in room B, the agent has to collect the ball with same color to the ball in room A. On the map, the red triangle is the agent, and light gray shows the area where agent can see. **Top Right:** Performance on Visual Match task. **Bottom:** The memories in FIFO and Place memory architectures.

## D.10 VISUAL MATCH TASK ON 2D MINIGRID

We evaluated SAT-AMA for the Visual Match task (Hung et al., 2019). It is similar to the task in the section 3.3 as illustrated in the top left figure of Figure 20, but in this environment, the agent should learn how to act from at the beginning of the episode where the agent observes one ball. As in the section 3.3, the memory capacity is a half of the number of steps in the second room, and PPO is used also. AMA design is also the same. The result is shown in the top right figure of Figure 20. As in the section 3.3, the SAT-AMA solves this task also, because the place memory can keep the memory at the beginning the episode by managing the FIFO memories per place as shown in the bottom figure of Figure 20.

## D.11 SAT-AMA IN HABITAT ENVIRONMENT

To further investigate the applicability of SAT-AMA for embodied agents in realistic 3D environment, we evaluated our model in Habitat environment (Savva et al., 2019). Habitat environment is a 3D simulator that allows the embodied agent to freely navigate in a house-like environment. We designed two tasks to demonstrate the applicability of SAT-AMA in this environment. The first task is a classification task, which requires the model to extract relevant patterns from complex scenes in order to perform well. The second task is a generation task, which requires the model to generate unseen scenes based on spatially-aware memory.

For classification task, we designed the task similar to RL task in Section 3.3 and Visual Match task in Appendix D.10. At the beginning of the episode, the agent is randomly spawned in the house, and the agent can observe the small colored square (one of red, blue, green, cyan, grey, yellow) randomly located in the scene (Figure 21(c)). Then, the square disappears and the agent randomly navigates in the house. After navigating 200 steps, the agent stops at a certain location and then randomly wander around nearby for another 200 steps. Thus, the agent observed total 400 frames in total. To this end, the first observation without the square is given as a query, and the model is asked to predict the color of the square. To acquire the place information, we calculated the distance between the given location and center location of each room, and chose its place to the closest room (Figure 21(a)).

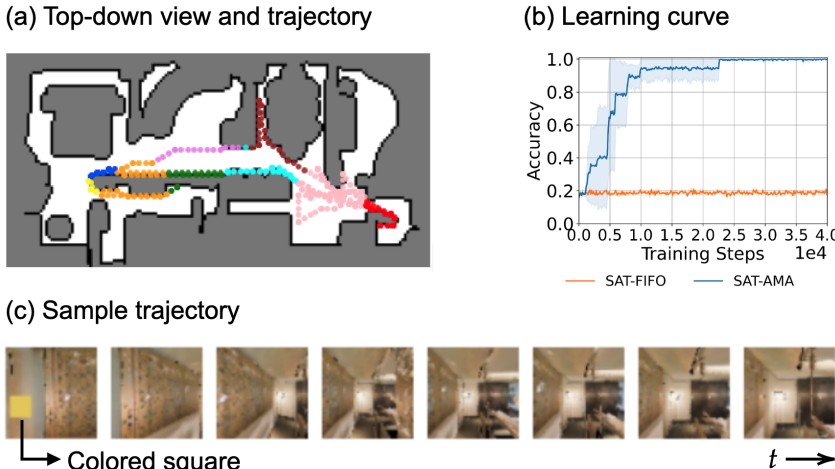

**Figure 21:** (a) Top-down view of the environment and agent's trajectory. Same colored points indicate same place. (b) Performance of SAT-FIFO and SAT-AMA on Habitat classification task. SAT-AMA learns to choose optimal strategy and attends to relevant memory in visually complex environment. (c) Sample trajectory of the episode. The agent observes the colored square at first step, then randomly navigates the house for 400 steps.

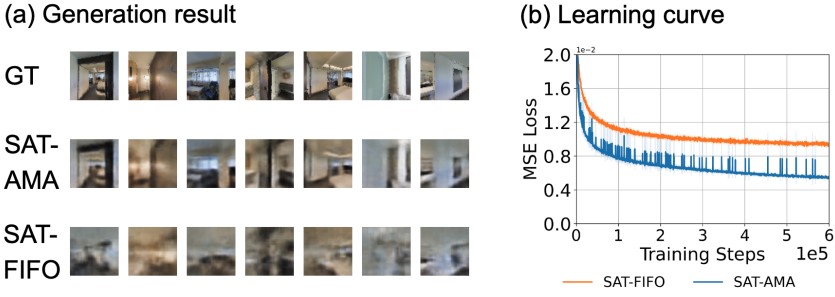

**Figure 22:** (a) Generation result of SAT-AMA and SAT-FIFO. Random unseen direction is given as a query, and SAT-AMA could generate the scenes well, while SAT-FIFO fails to. (b) MSE loss of SAT-FIFO and SAT-AMA.

To evaluate SAT-AMA and SAT-FIFO, we confined the memory capacity to 200. As illustrated in Figure 21(b), SAT-AMA can fully solve the task while SAT-FIFO performs near to random choice, about $18.75\%$ accuracy. SAT-FIFO's failure is because of the limitation of the memory capacity (it only can store the last 200 observations). Whereas, SAT-AMA can memorize the first observation by choosing MVFO strategy. This shows that SAT-AMA can extract the relevant patterns in complex scenes.

For generation task, we designed the task inspired by (Rosenbaum et al., 2018; Eslami et al., 2018). At the beginning of the episode, the agent is randomly spawned in the house, and the agent rotates to random direction repeatedly. After rotating for 150 steps (single rotation changes the direction by 12 degrees), the agent randomly rotates for another 150 steps within a range of 36 degrees in either direction from its current orientation. At the end, random unseen direction is given as a query, and the model is asked to generate the scene in given query direction. We use a fixed set of unseen directions for simplicity. To acquire the place information, we equally divided $360°$ into 10 regions, and assumed each region as a place.

To evaluate SAT-AMA and SAT-FIFO, we limited the memory capacity to 150. As illustrated in Figure 22(b), SAT-AMA outperforms SAT-FIFO in reconstruction loss. This is because SAT-FIFO stores last 150 observations which cover only some portion of the entire $360°$ scene. However, SAT-AMA can choose the MVFO strategy to store entire $360°$ scene effectively attending to relevant scenes for generation. The visualizations in Figure 22(a) shows that SAT-AMA could generate the

scenes well, while SAT-FIFO fails to. This implies that SAT-AMA can generate unseen scenes based on spatially-aware memory.

| Tasks | Descriptions |
|---|---|
| LIFO | Recall the first 9 performers out of the 18 provided. |
| | Think back to the initial half of the 18 dancers. |
| | Keep the first nine from the 18 dancers in mind. |
| | Don't forget the opening 9 out of the presented 18. |
| | Reflect on the earliest set of nine in the group of 18. |
| | Remember the starting group among the specified 18 dancers. |
| | Consider the first batch from the total 18 dancers. |
| | Hold onto memories of the leading 9 out of the 18. |
| | Bring to mind the first subset from the given 18 performers. |
| | Think of the primary nine in the group of 18. |
| | Refresh your memory about the starting nine of the 18 dancers. |
| | Jog your memory concerning the first half of the specified 18. |
| | Retain thoughts of the premier nine from the ensemble of 18. |
| | Retrospect on the foremost 9 dancers among the 18. |
| | Focus on the first cluster of nine from the total 18. |
| | Ponder the initial dancers in the lineup of 18. |
| | Recollect the opening segment of nine from the presented 18. |
| | Keep in your thoughts the primary set of dancers in the group of 18. |
| | Consider the inaugural nine performers from the indicated 18. |
| | Remember the principal 9 dancers from the ensemble of 18. |
| FIFO | Think of the final nine from the group of 18 dancers. |
| | Recall the concluding nine performers of the 18. |
| | Bring to mind the last group among the 18 dancers. |
| | Reflect on the 18 dancers particularly the last nine. |
| | Keep in mind the nine dancers who concluded the group of 18. |
| | Remember the dancers who came last in the set of 18. |
| | Consider the final segment of dancers in the 18. |
| | Focus on the last nine of the total 18 dancers. |
| | Who were the concluding performers in our set of 18? |
| | Cast your thoughts to the tail end of the 18 dancers. |
| | Think back to the 18 dancers especially the concluding nine. |
| | Recall the tail end of the 18-dancer lineup. |
| | Retain memories of the nine dancers wrapping up the list of 18. |
| | Mull over the dancers that concluded our 18-person roster. |
| | Bring forward the memory of the concluding acts among the 18. |
| | Remember the last stretch of dancers in the 18. |
| | Who rounded off the 18-dancer performance? |
| | Reminisce about the final cluster among the 18 dancers. |
| | Keep the concluding dancers in focus from the group of 18. |
| | Recapture the last performers from the total of 18 dancers. |

**Table 4:** The task descriptions for LIFO and FIFO tasks.

| Tasks | Descriptions |
|---|---|
| LVFO | Recall the dancers in the room the agent frequented the most.
Think of the dancers in the room that the agent visited most often.
Bring to mind the performers in the agent's most-visited room.
Remember the dancers from the room where the agent spent the most time.
Reflect on the dancers in the room that seemed to be the agent's favorite.
Consider the dancers in the chamber the agent went to repeatedly.
Who were the dancers in the room the agent kept returning to?
Jog your memory about the dancers in the most visited room by the agent.
Ponder on the performers in the room the agent seemed to prefer.
Retrospect on the dancers in the space the agent was drawn to the most.
Keep in mind the dancers from the room the agent visited most frequently.
Recollect the dancers in the room that caught the agent's attention the most.
Focus on the dancers from the room the agent seemed most interested in.
Retain memories of the performers where the agent made the most stops.
Mull over the dancers in the room the agent returned to time and again.
Reflect upon the dancers from the agent's most frequented chamber.
Dwell on the performers in the room the agent seemed most attached to.
Hold the thought of the dancers in the room with the highest visits by the agent.
Contemplate the dancers in the room the agent seemed to gravitate towards the most.
Remember the performers in the space the agent appeared to favor. |
| MVFO | Recall the most recent dancers from every room.
Think of the dancers who performed last in each space.
Bring to mind the final performers in every chamber.
Reflect on the concluding dancers from all rooms.
Consider the last dancers to grace each stage within the rooms.
Who were the concluding performers in each of the rooms?
Ponder on the dancers who wrapped up the acts in every room.
Jog your memory about the closing performers from each chamber.
Recollect the tail-end dancers in every room.
Focus on the final acts of dancers in all the rooms.
Retain memories of the performers who closed the show in each space.
Mull over the dancers who rounded off the performances in every chamber.
Contemplate the last artists to take the stage in each of the rooms.
Think back to the closing acts from every performance space.
Zero in on the concluding dancers in all the chambers.
Dwell upon the artists who were last to perform in every room.
Cast your mind to the end performers in each of the spaces.
Reflect upon the dancers who ended the sequence in each chamber.
Reminisce about the performers who took the final bow in all rooms.
Remember the artists who concluded the dances in every individual space. |

**Table 5:** The task descriptions for LVFO and MVFO tasks.

