# OpenReview forum: "Spatially-Aware Transformers for Embodied Agents"
_ICLR.cc/2024/Conference — ICLR 2024 spotlight_

### Official Review · Reviewer_44cr · 2023-10-22

**Soundness:** 3 good
**Presentation:** 3 good
**Contribution:** 3 good
**Rating:** 6
**Confidence:** 4

**Summary:**

The paper proposes Spatially-Aware Transformers (SAT) to keep spatial information in place-centric episodic memory.
SAT maintains multiple spatial memories for respective places while ensuring the "total" size of memory (i.e., $L / K$).
For memory management, the author proposes the Adaptive Memory Allocator (AMA) that adaptively finds the best memory management policy by learning $\pi(\sigma | \tau)$ using Q-learning given a task description, $\tau$.
The proposed AMA outperforms the baselines (i.e., model w/o AMA) by noticeable margins regarding effectiveness and efficiency in various downstream tasks.

**Strengths:**

- The paper is generally written well and easy to follow.
- Extending Transformer-based memory architecture to spatial domains is well-motivated and sounds sensible.
- Learning to choose the best memory management strategy from multiple candidates looks reasonable.
- Experiments on various downstream tasks supports the generality of the proposed approach.
- The proposed approach achieves strong performance gain with large margins.

**Weaknesses:**

- The novelty of AMA seems a bit weak, as it is basically policy learning that chooses the best action (here, strategy) that maximizes rewards. What are some core differences from conventional policy learning, especially related to memory management for spatial information?
- The key idea for SAT seems to use separate networks for respective places. But can the architecture be useful in the case of a large number of places (i.e. what if $K -> \infty$ that results in almost zero size of memory for each place)?

**Questions:**

See weaknesses above.

---

> ### Author Response · Authors · 2023-11-23
>
> We are grateful to Reviewer 44cr for their thoughtful critique and acknowledgment of the well-written nature of our manuscript and its contributions to spatial memory in transformer architectures. Your feedback has been invaluable in enhancing the quality of our paper.
>
> ### **Novelty of the Adaptive Memory Allocator (AMA)**
>
> We answered this in Common Response CR3.
>
> ### **Scalability with a Large Number of Places**
>
> Regarding scalability, our Spatially-Aware Transformer (SAT) architecture is designed with parameter sharing across place memories to encode observations efficiently. By employing a common network for all place memories, the SAT model can effectively attend to and process information pertinent to each specific place. This architecture enables the transformer to selectively retrieve relevant memory in response to queries, regardless of the number of places. The number of place $K$ could not be infinite, the maximum number of place is the number of memory, in which, the Place Memory will be same to SAT memory without the hierarchy.

---

> > ### Comment · Reviewer_44cr · 2023-12-04
> > **Official Comment by Reviewer 44cr**
> >
> > I thank the authors for addressing my concerns. The provided response cleared up my raised concerns and thus, I would like to keep my original positive rating.

---

### Official Review · Reviewer_QvRD · 2023-10-28

**Soundness:** 4 excellent
**Presentation:** 4 excellent
**Contribution:** 3 good
**Rating:** 8
**Confidence:** 4

**Summary:**

The paper investigates the spatially-aware transformers (SAT) that incorporate agent's spatial and temporal experiences important to solve embodied problems. Specifically, it studies the influence of spatial and decisional information, and explores multiple memory management strategies (e.g., Place-centric vs Time-centric, First-In-First-Out vs Last-In-First-Out, and Adaptive Memory Allocator) on various downstream tasks, showing insights of different memory utilization and management approaches in addressing many machine learning problems (e.g., supervised prediction, image generation), and justifying the proposed SAT-AMA.

**Strengths:**

- This paper studies the important problem of utilizing and managing spatial and temporal memory experienced by embodied agents that are essential to address various downstream tasks. Particularly, it considers the practical issue of memory constraint and perform experiments based on the popular transformer architectures. The research presented in this paper is very well motivated.

- The paper is very technical solid. Almost all arguments are well-supported/justified by the highly-relevant and classic references and thorough experiments (in Appendix). It carefully compares place- and time-centric store and read, and progressively studies FIFO to the proposed Adaptive Memory Allocator based on SAT. I believe the extensive settings and results presented in this paper have the potential to inspire many future works.

- Overall, this paper was a very enjoyable read to me. All details have been clearly presented (especially with the Appendix and all nice visualizations); The paper is nicely-structured, concise but contains massive valuable information. I believe many arguments and thoughts presented in this paper will be very constructive to relevant future research.

**Weaknesses:**

- The title of this paper is very misleading
    - The paper does not propose any new formation of the transformer architecture to better model spatial information, but focuses on managing agents' episodic memory for addressing different downstream tasks.

- This paper overclaims several contributions.
    - The paper says "we are the first to motivate, conceptualize, and introduce the notion of transformers capable of utilizing explicit spatial information", but as the authors mentioned that defining and managing external memory have been extensively studied in previous machine learning literatures. It simply compares place- and time-centric store and hierarchical read methods that are intuitively suitable for different downstream tasks.
    - One good example is the use of topological graph that stores observations of keypoints for agents in visual navigation, e.g., the widely applied DUET agent [1] for vision-and-language navigation [2], which is essentially close to the proposed SAT-PM-PH model.
    - The experiments of action-conditioned image generation (Exp-5) and reinforcement learning agents (3.3) are not very convincing to me. Exp-5 is not a practical setting and 3.3 is relatively simple that cannot represent other reinforcement learning agents, see more below.

- Missing experiments.
    - I think this paper lacks investigation on the more recent embodied agents and their memory management approaches.
    - The experiments presented in this paper are relatively small-scale and simple. I am concerning how the proposed methods might impact recent research that often apply more capable networks (and massive data) to learn generic spatiotemporal priors to facilitate representing and memorizing the observations. e.g., Figure 3 - an agent might be able to create a very compact representation for all ballet rooms from a single visit to each room?
    - Following my previous point, I think this study emphasizes the scenario of memory constraint, but the experiments only considers small memory capacity, negelating the difference in representation (i.e., how to store an observation) and the difficulty in grounding from the queries to relevant memories. Overall, the generalization and practical influence of the proposed methods is not very clear to me.
    - I think a valuable baseline is missing here: without explicitly defining any memory management strategies but gives the agent a certain memory budget and asks it to learn to update the memory by itself by training the agent on a mixture of data and tasks (e.g., a more general version of DNC mentioned in Appendix D.4).

[1] Think Global, Act Local: Dual-scale Graph Transformer for Vision-and-Language Navigation. Chen et al., CVPR2022.

[2] Vision-and-Language Navigation: Interpreting Visually-Grounded Navigation Instructions in Real Environments. Anderson et al., CVPR2018.

**Questions:**

I hope the authors can address some of my concerns in Weaknesses. I don't have any other questions here.

---

> ### Author Response · Authors · 2023-11-23
>
> We would like to extend our sincere thanks to Reviewer QvRD for their positive and encouraging feedback on our manuscript. We are grateful for the recognition of the importance of our research and the comprehensive nature of our work. Your constructive comments have been instrumental in refining our paper.
>
> ### **Concerns on Misleading Title**
>
> Your observation regarding the title's potential to mislead readers is well-noted. We added 'Memory' in the title.
>
> ### **Concerns on Over-claiming**
>
> >The paper says "we are the first to motivate, conceptualize, and introduce the notion of transformers capable of utilizing explicit spatial information", but as the authors mentioned that defining and managing external memory have been extensively studied in previous machine learning literatures. One good example is the use of topological graph that stores observations of keypoints for agents in visual navigation, e.g., the widely applied DUET agent [1] for vision-and-language navigation [2], which is essentially close to the proposed SAT-PM-PH model.
> >
>
> We acknowledge the similarities between our SAT-PM-PH model and the topological map based memory used in the DUET agent for vision-language navigation. However, our work extends beyond the storage of keypoint observations by embedding the spatial knowledge within the transformer’s attention mechanism, which allows for a more nuanced and integrated processing of spatial information. For example, the place memory could be constructed per place, the each memory is encoded with more detailed spatial information. Other differences are that our model is designed to be retrieved without the spatial information and to optimize the efficacy on the retrieval through the hierarchical architecture, while the topological map based memory doesn’t.
>
> In light of your feedback, we will revise our manuscript to more accurately represent the state of the art and clearly delineate the innovative aspects of our work. We will adjust our language to avoid overstatements and ensure that our contributions are contextualized appropriately within the broader landscape of research in this area.
>
> > The experiments of action-conditioned image generation (Exp-5) and reinforcement learning agents (3.3) are not very convincing to me. Exp-5 is not a practical setting and 3.3 is relatively simple that cannot represent other reinforcement learning agents, see more below.
>
> To handle this, we additionally evaluated our model in realistic 3D environment, Habitat environment (suggested by reviewer XL1m). See Common Response CR2 for more details.
>
> ### **Missing Experiments**
>
> > I think this paper lacks investigation on the more recent embodied agents and their memory management approaches.
> >
>
> We thank the reviewer for this constructive feedback. We acknowledge that we overlooked some recent embodied agents such as DUET agent. We will update the experiments with the more recent embodied agents in our revision.
>
> > The experiments presented in this paper are relatively small-scale and simple. I am concerning how the proposed methods might impact recent research that often apply more capable networks (and massive data) to learn generic spatiotemporal priors to facilitate representing and memorizing the observations. e.g., Figure 3 - an agent might be able to create a very compact representation for all ballet rooms from a single visit to each room?
> >
>
> We appreciate your constructive feedback. As we replied for your concern above, we additionally evaluated our model in the more realistic 3D environment, Habitat environment, which is discussed in our common responses.
>
> > Following my previous point, I think this study emphasizes the scenario of memory constraint, but the experiments only considers small memory capacity, neglating the difference in representation (i.e., how to store an observation) and the difficulty in grounding from the queries to relevant memories. Overall, the generalization and practical influence of the proposed methods is not very clear to me.
> >
>
>  We are grateful for your critical insight regarding our treatment of memory storage strategy and retrieval of relevant memories from queries.
>
>  We recognize that the memory storage strategy based on observation differences represents a valuable line of inquiry [1, 2]. As mentioned in our common responses, our primary research question in this paper is “what if spatial information is available as time indices to transformer-based episodic memory?”. While the memory storage strategy based on observation difference is indeed an important aspect, it is somewhat orthogonal to the main focus of our study. Nevertheless, we agree that such a strategy could be a robust alternative to FIFO memory and could complement our spatially-aware memory framework.

---

> ### Author Response · Authors · 2023-11-23
>
> For retrieving the relevant memories, we think that it is shown in **Exp-5**. In the experiment, the model should generate the mini-patch on the facial image even though sometimes, the appearance looks similar (e.g., when the patch is near to the eye). Figure 7 shows that SAT-AMA can generate the mini-patches accurately based on the actions and spatial knowledge. This result suggests that SAT-AMA can effectively discern relevant memories through the spatial knowledge even when observation differences are subtle.
>
> We thank the reviewer for giving this insightful comment. We will try to add this strategy as one of our baselines in our revision.
>
> >I think a valuable baseline is missing here: without explicitly defining any memory management strategies but gives the agent a certain memory budget and asks it to learn to update the memory by itself by training the agent on a mixture of data and tasks (e.g., a more general version of DNC mentioned in Appendix D.4).
>
> We appreciate your suggestion to evaluate the Differentiable Neural Computer (DNC) across a mix of tasks. Even though we evaluated DNC for a single task in the Room Ballet environment, but we extended this experiment to cover the setting with multiple tasks. During this process, we observed that the memory capacity allocated to the DNC in prior experiments was less than that of the SAT-AMA. To address this discrepancy, we reassessed the DNC performance on a single task with a memory capacity now comparable to that of SAT-AMA.
>
> These results are detailed in **Appendix D.6**. In summary, when memory capacities are matched, the DNC demonstrates quicker learning than SAT-AMA in terms of training steps. However, it is notably less efficient in terms of wall-clock time, and the performance is unstable throughout learning process. The slow learning speed in terms of wall-clock time is attributed to the recurrent nature of the DNC's architecture, which, unlike the parallel training capabilities of the Transformer, necessitates a slower training process.
>
> Furthermore, the AMA's policy requires a period of exploration during the initial stages of learning, which may contribute to a comparative delay in learning speed when measured by training steps. These insights and additional details are expounded upon in the Appendix. We are grateful for your constructive feedback.
>
> [1] Le, Hung, et al. "Intrinsic Motivation via Surprise Memory." *arXiv preprint arXiv:2308.04836* (2023).
>
> [2] Saade, Alaa, et al. "Robust Exploration via Clustering-based Online Density Estimation." (2022).

---

### Official Review · Reviewer_XL1m · 2023-10-29

**Soundness:** 2 fair
**Presentation:** 2 fair
**Contribution:** 3 good
**Rating:** 6
**Confidence:** 3

**Summary:**

Inspired by the significance of spatial context in the formation and retrieval of episodic memory, the paper proposes to add a spatial embedding into transformers and organize the episodic memory in a place-centric way to obtain a spatially aware Transformer model. The paper also proposes the Adaptive Memory Allocator, a memory management method based on reinforcement learning that aims to optimize the efficiency of memory utilization. The experiments on several environments demonstrate the advantages of the proposed model.

**Strengths:**

- It's exciting and of significance to have a transformer capable of utilizing explicit spatial information and can act as better episodic memory.

- Designing an adaptive memory allocator is useful

- Extensive experiments on various environments and tasks.

- The idea of incorporating spatial information into episodic memory is novel and interesting

**Weaknesses:**

- It seems the "spatial-aware transformers" are achieved solely by adding a spatial embedding. And its specific design or implementation is not clearly mentioned. Only a sinusoidal positional embedding is mentioned in Exp-1, do other experiments also use this? How is this positional embedding enough to represent spatial relations, since space is not unidirectional as time?

- The idea of an ADAPTIVE MEMORY ALLOCATOR is very interesting, but the implementation is quite trivial to me. What's the difficulty of learning this policy through Q-learning?

- The organization of the paper could use some improvements, such as the missing Table 1 in the paper though mentioned in Exp-2, the confusing combination of the figures from different experiments, and many details deferred to the appendix, making it difficult to understand the details of the model or the experiments.

- Much more details of the experiments are needed. What are the exact input and output of these environments? How is the input represented and fed into the transformers? How is the training implemented for different baselines?

- The experiment environments are too toy setting for "embodied agents", it would strengthen the paper to add some real embodied environments, such as the Habitat[1] and TDW[2].

[1] Habitat: A Platform for Embodied AI Research

[2] ThreeDWorld: A Platform for Interactive Multi-Modal Physical Simulation

**Questions:**

See concerns in the weaknesses.

---

> ### Author Response · Authors · 2023-11-23
>
> We express our gratitude to Reviewer XL1m for their insightful feedback and for recognizing the potential of our work to enhance episodic memory with spatial context. We are particularly appreciative of the acknowledgment of the significance of our proposed Adaptive Memory Allocator (AMA) and the comprehensiveness of our experimental validation.
>
> ### **Clarification on the Design and Implementation of Spatial Embeddings**
>
> Thank you for the constructive suggestion. We agree that our paper would become more clear and complete with the suggested improvement. As suggested, we made the following improvements.
>
> In **Section 2.1**, we added the following.
>
> “For the time and spatial embedding, the learnable embedding \citep{bert} or sinusoidal positional embedding \citep{transformer} could be applicable, but in this literature, for simplicity, the sinusoidal positional embedding is used.”
>
> In **Appendix C.2**, we provided a comprehensive description of the spatial embedding construction for each experiment. For example, we clarified the fact that we used 2-D sinusoidal positional embedding in **Exp-5** (the FFHQ generation experiment) while sinusoidal positional embeddings were primarily employed in other experiments.
>
> In **Appendix D.9**, we added a comparative analysis of sinusoidal, 2-D sinusoidal (sinusoidal embedding for x and y axes), and 2-D learnable positional embeddings. This suggests that that 2-D sinusoidal and learnable embeddings are comparable while the sinusoidal embedding is clearly worse than them.
>
> ### **What’s the difficulty of learning AMA policy**
>
> We answered this in Common Response CR3.
>
> ### **Improving the Organization of the Paper**
>
> Thank you for the constructive suggestion. We agree with the potential confusions and need for update. Unfortunately, due to the page limit, we are currently facing difficulties in incorporating certain Figures and Tables from the Appendix into the main paper. We will, however, continue exploring options to appropriately arrange the content. For instance, if the paper is accepted, it would be much easier to make the suggested updates as we would be allowed an additional page. In the meantime, we have clearly indicated in the paper that Table 1 is located in the Appendix.
>
> ### ****Application to the more realistic embodied environments****
>
> Thank you for the suggestion. As suggested, we conducted additional experiment in a complex environment Habitat. See Common Response CR2 for more details.
>
> ### **Details of the experiments are missing. Input & Output of each environment, and training algorithms for different baselines**
>
> Thank you for the suggestion. As per your advice, we have included the suggested details of the experiments in **Appendix C.1**. These details include the exact input and output of each environment, as well as the representations that are fed into transformer memories.
>
> Regarding the training algorithm of the baselines, we have provided the details in **Appendix B.3**. The training algorithm for SAT-FIFO is the same as SAT-AMA, except that it does not require learning to choose a strategy and therefore does not involve policy learning. In the case of DNC, we sampled a batch of episodes and directly optimized the downstream task loss function.
>
> The detail models and hyperparameters of SAT variants and DNC are provided in **Appendix B.4** and **Appendix D.6**, respectively. For reproducibility, we will also release the code upon the acceptance of the paper.

---

> > ### Comment · Reviewer_XL1m · 2023-12-04
> >
> > Thanks for the detailed response and for updating the paper! Most of my concerns have been addressed. After reading all the reviews and responses, I decided to raise my score to 6, leaning towards acceptance.

---

### Official Review · Reviewer_BEnq · 2023-11-01

**Soundness:** 3 good
**Presentation:** 3 good
**Contribution:** 4 excellent
**Rating:** 8
**Confidence:** 3

**Summary:**

The paper introduces and evaluates a novel framework comprising the Spatially-Aware Transformer (SAT) and Adaptive Memory Allocator (AMA) for a range of tasks demanding spatial awareness and effective memory management. Through a series of well-structured experiments, the authors demonstrate the capabilities of their proposed methods in various environments, such as the Room Ballet environment for prediction tasks, as well as in action-conditioned world modeling and spatially-aware image generation. The comprehensiveness of the methodology is evident, as it details how to incorporate spatial information into transformer models and effectively manage memory for different types of tasks. The experiments are extensive and cover different scenarios to validate the efficacy of the proposed framework.

**Strengths:**

**Great Motivation:** The paper addresses a critical gap in existing models’ inability to effectively integrate spatial information, which is very crucial for tasks in various domains. The introduction and literature review provides a compelling argument for why this integration is necessary, setting a solid foundation for the rest of the paper.

**Comprehensive Method Explanation:** The authors provide a thorough and clear explanation of the Spatially-Aware Transformer and Adaptive Memory Allocator. The methodology section is well-structured, detailing each component of the system, the underlying theory, and the implementation specifics, which aids in the reproducibility of the results.

**Extensive Experiments:** The paper goes beyond theoretical claims and validates the proposed framework through a series of diverse and challenging experiments. These experiments not only demonstrate the strengths of the SAT-AMA combination but also highlight its versatility across different tasks and scenarios. The image generation experiments are especially interesting and seem not to exist in its ancestor work of Towards mental time travel: a hierarchical memory for reinforcement learning agents.

**Weaknesses:**

**Usability in Complex Embodied AI Scenarios:** The paper, while comprehensive, could benefit from a deeper discussion on the applicability and scalability of the proposed methods in more complex embodied AI scenarios. Given the rising interest in virtual homes and ThreeDWorld with many rooms for task operation, readers would appreciate some discussion or insights into how the SAT-AMA framework can be adapted or scaled to meet the challenges presented by these intricate environments. A similar discussion could be like, "Is the method scalable to larger environments?" and "How can the SAT-AMA framework be adapted for more intricate task operations?" to provide a complete picture to the readers.

**More discussion on AMA:** I appreciate the introduction and application of the Adaptive Memory Allocator (AMA) within the Spatial Awareness Transformer framework, as it presents a novel and promising approach to dynamically allocate memory based on task requirements. However, I find that there could be a more detailed analysis and discussion of AMA’s strategy selection across different experimental settings and tasks. Understanding how AMA decides on specific spatial strategies could uncover valuable priors for selecting appropriate strategies based on the nature of the task, which would immensely benefit future work exploring new methods in this domain. It would be beneficial for the readers if the authors could provide insights into the distribution of strategies chosen by AMA, its dependency on the nature of tasks, and the correlation between strategy choices and task performance. Such analysis would not only deepen our understanding of AMA’s workings but also offer practical guidance for researchers aiming to employ similar adaptive mechanisms in their work. A discussion on these aspects would significantly enhance the completeness and depth of the paper, providing valuable context and potentially guiding future innovations in the field.

**Questions:**

See above

---

> ### Author Response · Authors · 2023-11-23
>
> We express our gratitude to the reviewer for the recognition of our paper's motivation and the breadth of our experiments. As addressed in the common responses, we have discussed the extension of our work to the complex embodied AI experiment in the Habitat environment.
>
> ### **Usability in more complex embodied AI scenarios**
>
> As suggested, we added additional experiment results in a complex environment Habitat (suggested by reviewer XL1m). See Common Response CR2 for more details.
>
> ### **More discussion on AMA**
>
> We thank the reviewer for their insightful comments regarding the need for more in-depth discussion on the Adaptive Memory Allocator (AMA). We agree that providing a more comprehensive discussion on AMA would significantly enhance the paper.
>
> As suggested, we investigated further to show the distribution of strategies chosen by AMA, and discuss its dependency on the nature of tasks, and the correlation between strategy choices and task performance. The table below presents the distribution of strategies selected by AMA, along with corresponding performance metrics, derived from the Ballet-FIFO task in **Exp-3**. We note that each column represents the probability of each strategy being selected, the accuracy when each strategy was selected. The last row represents the accuracies normalized for the four strategies.
>
> The strategy distribution was averaged across three distinct runs (random seeds) and was ascertained by passing the output of the AMA's policy network through SoftMax. The results elucidate a clear correlation between the chosen strategies and task performance, reinforcing the premise that AMA's strategic choices are inherently linked to efficacy.
>
> |  | FIFO | LIFO | MVFO | LVFO |
> | --- | --- | --- | --- | --- |
> | AMA dist. | 0.475 | 0.073 | 0.15 | 0.302 |
> | Accuracy | 0.999 | 0.146 | 0.325 | 0.714 |
> | Normalized Acc. | 0.457 | 0.067 | 0.149 | 0.327 |
>
> We added this new results in **Appendix D.4** of the revised version.
>
> Again, we appreciate your suggestion. We believe that by incorporating this analysis, the paper will become more comprehensive.

---

### Author Response · Authors · 2023-11-23
**Common Responses**

We appreciate the reviewers for the constructive feedbacks. We revised our paper by considering the comments, especially they are highlighted with the colors which are **blue** for **BEnq**, **magenta** for **XL1m**, **red** for **QvRD**, and **cyan** for **44cr**.

### **CR1. Strengths**

We sincerely thank the reviewers for their insightful comments and constructive feedback. Reviewer BEnq described the paper as having “great motivation”, and being “very crucial” with a “thorough and clear explanation”. The paper is also highlighted with “extensive experiments” and “diverse environments” from reviewer BEnq, and XL1m. The reviewers also noted the paper's strong technical foundation, with reviewer QvRD describing it as "very technical(ly) solid" and containing "almost all arguments are well-supported." They also emphasized the paper's potential to inspire future research and its enjoyable and concise structure. Reviewer 44cr praised the paper for being well-written, easy to follow, and well-motivated.

### **CR2. Experiments on Realistic 3D Environment**

We are grateful for the insights provided by reviewers BEnq, XL1m, and QvRD on the applicability and scalability of our methods in complex embodied agent scenarios. While the core contribution of our study is the novel conceptualization of spatially-aware structured transformer memory and the demonstration of its benefits in various applications, we acknowledge the importance of extending this applicability to more intricate and realistic settings. As suggested, we therefore, conducted additional experiments in more realistic 3D environment, the Habitat environment [2], and updated the paper with the additional results in **Appendix D.11**.

Specifically, we designed two tasks, classification and generation. The classification task evaluates the model's ability to extract relevant patterns in complex scenes. In each episode, the agent observes a small colored square in the first observation, which then disappears. The agent navigates randomly throughout the house, eventually stopping at a specific location. It then continues to wander nearby during another phase. The model is then given the first observation without the square and asked to predict the color of the square observed at the beginning of the episode. We compared SAT-AMA with SAT-FIFO, and the results are shown in the table below. Please note that due to time constraints, we were unable to conduct additional experiments on more baselines. However, we will include more complete results in the camera ready version.

|  | SAT-AMA | SAT-FIFO |
| --- | --- | --- |
| Accuracy (%) | 99.79 | 18.75 |

In the generation task, the task is to generate unseen scenes based on spatially-aware memory. Each episode starts with the agent being randomly spawned in a house and rotating in random directions. After a certain number of rotations, the agent is given a random unseen direction as a query and is asked to generate the scene in that direction. The results are shown in the following table.

|  | SAT-AMA | SAT-FIFO |
| --- | --- | --- |
| MSE loss | 5.572e-3 | 9.235e-3 |

### **CR3. Novelty of the Adaptive Memory Allocator (AMA)**

We acknowledge that the implementation of AMA may initially appear as a simple one-step Q-learning process. However, the novelty of AMA lies in its incorporation of strategies and their consideration as actions. This is crucial because using a standard RL policy without strategies for memory management would require selecting a specific memory position as an action, resulting in scalability issues. In other words, training becomes increasingly challenging as the memory size grows. By introducing the concept of strategy, which is designed by a human, our paper demonstrates that we can address this issue and ensure scalability. Nevertheless, this approach does have a drawback. Its main limitation is that the strategy is designed rather than learned. An interesting direction for future research would be to develop an approach that can learn the strategy while still maintaining scalability.

[1] Graves, Alex, et al. "Hybrid computing using a neural network with dynamic external memory." Nature 538.7626 (2016): 471-476.

[2] Savva, Manolis, et al. "Habitat: A platform for embodied ai research." *Proceedings of the IEEE/CVF international conference on computer vision*. 2019.

---

> ### Author Response · Authors · 2023-11-23
> **Revisions in the updated manuscript**
>
> - To address the reviewer XL1m’s concern for clarifying the spatial embedding,
>     - In Section 2.1, we added the detailed description for which embedding is used for the spatial embedding.
>     - In **Exp-5**, we added the additional explanation of used spatial embedding for this experiment.
>     - We added **Appendix C.2** to give the detailed information of the utilized spatial embedding for each task.
> - To address the reviewer XL1m’s concern for a lack of the detailed information of the experiments,
>     - In Section 3, we added the additional explanation about the experimental setting.
>     - We added **Appendix C.1** to share more detailed information about the input and expected output for each task.
>     - We updated **Appendix B.3.** to share the training algorithms of different baselines.
> - To address the reviewer BEnq’s request for more discussion on AMA,
>     - We added **Appendix D.4** to investigate the correlation between the chosen strategies distribution and task performances.
> - To address the reviewer QvRD’s request to investigate further for the learning-based read/write mechanism,
>     - We updated **Appendix D.6** to evaluate the Differentiable Neural Computer (DNC) [1] compared to our model for multi-task setting.
> - To address the reviewers BEnq, XL1m, and QvRD’s concern for applicability and scalability in more realistic environment,
>     - We added **Appendix D.11** to evaluate the our model in the more realistic 3D environment, Habitat environment.

---

### Meta-Review · Area_Chair_Ur2F · 2023-12-05

**Metareview:**

Overall, all reviewers (and this AC) were unanimously positive about this paper after the rebuttal/discussion process. Collectively, we found it to be well-motivated, novel, interesting and clearly presented. The paper is solid (esp. after revisions), with demonstrated generality and good results.

That said, in the initial reviews, reviewers pointed out valid major concerns about needing experiments with more complex environments (e.g. Habitat, ThreeDWorld, etc), novelty/triviality of the AMA implementation, and some overclaiming in the title and paper.

However, the authors added experiments with Habitat, provided additional analyses and explanations/justifications, while also revising the title and claims. Ultimately, some reviewers revised their scores upwards, and all reviewers found their major concerns to be addressed satisfactorily.

In summary, this is a nice paper that will certainly be interesting for the ICLR community. In particular, this AC very much likes the novel angle/motivation of spatial episodic memory. I think this paper will move the embodied AI field forward in interesting new ways, and it's certainly more exciting beyond achieving SOTA performance on existing tasks.

**Justification For Why Not Higher Score:**

While I really like this paper, it would have been stronger if it had included results on Habitat, ThreeDWorld, etc. from the start. Instead, the authors provided results only on Habitat during the rebuttal discussion process.

Perhaps it wouldn't have been possible to squeeze all of the new problem framing, various architectures and all the ideal experiments+results in an ICLR paper length, but unfortunately for me, this paper falls short of an oral standard primarily because it was only tested on one standard complex embodied environment (i.e. Habitat).

**Justification For Why Not Lower Score:**

As described in the meta-review, the paper is a good one on most or all counts: motivation, novelty, evaluation, presentation, etc.

I recommend it as a spotlight to allow greater promotion of such genuinely novel work (not easy to come by) that is also well-executed.

---

### Decision · Program_Chairs · 2024-01-16

Accept (spotlight)